# On Inductive Biases for Heterogeneous Treatment Effect Estimation

**Alicia Curth**
University of Cambridge
amc253@cam.ac.uk

**Mihaela van der Schaar**
University of Cambridge
University of California, Los Angeles
The Alan Turing Institute
mv472@cam.ac.uk

## Abstract

We investigate how to exploit structural similarities of an individual's potential outcomes (POs) under different treatments to obtain better estimates of conditional average treatment effects in finite samples. Especially when it is unknown whether a treatment has an effect at all, it is natural to hypothesize that the POs are similar – yet, some existing strategies for treatment effect estimation employ regularization schemes that implicitly encourage heterogeneity even when it does not exist and fail to fully make use of shared structure. In this paper, we investigate and compare three end-to-end learning strategies to overcome this problem – based on regularization, reparametrization and a flexible multi-task architecture – each encoding *inductive bias* favoring shared behavior across POs. To build understanding of their relative strengths, we implement all strategies using neural networks and conduct a wide range of semi-synthetic experiments. We observe that all three approaches can lead to substantial improvements upon numerous baselines and gain insight into performance differences across various experimental settings.

## 1 Introduction

The advent of fields such as personalized medicine has led to rapid growth of the machine learning (ML) literature on heterogeneous treatment effect estimation in recent years [1, 2, 3, 4, 5, 6, 7, 8]. To further advance the understanding of how to incorporate insights from other areas of ML into treatment effect estimation, we revisit the well-established problem of estimating the conditional average treatment effect (CATE) of a binary treatment within the potential outcomes (PO) framework [9]. The PO framework allows to conceptualize the problem as estimating the expected difference between an individual's expected 'potential' outcome with and without treatment, of which only one is observed in the *factual* world. This *fundamental problem of causal inference* [10] leads to the consensus that CATE estimation is not 'just another' supervised learning problem [6].

Under the standard assumption of ignorability – which precludes hidden confounding – we consider two *statistical* features central to estimating CATE: (i) the presence of confounding and (ii) CATE being a *contrast* between two PO functions, possibly exhibiting *simpler* structure than each PO separately. Much of the recent ML literature on CATE estimation has focused on the first feature, and treated confounding as a *covariate shift* problem. At this point, a range of sophisticated solutions exist which reduce the effect of confounding by balancing the covariate space [3, 4], importance weighting [11, 12, 13, 14] or propensity drop-out [15]. How to exploit the second feature in an *end-to-end manner*, however, has received little explicit attention so far and is what we focus on here.

We build on the intuition that the two tasks in the CATE problem – estimating the expected PO with and without treatment – are expected to be strongly related in practical applications (regardless of the

35th Conference on Neural Information Processing Systems (NeurIPS 2021).

presence of confounding). In fact, under the common null hypothesis of no treatment effect, we expect them to be identical. Even when there is a non-zero treatment effect, we expect shared structure: In medicine, for example, one distinguishes between biomarkers that are *prognostic* of outcome regardless of treatment status – hence determining what is shared between the POs – and biomarkers that are *predictive* of treatment effectiveness – determining heterogeneity [16, 17]. The induced difference between the POs is

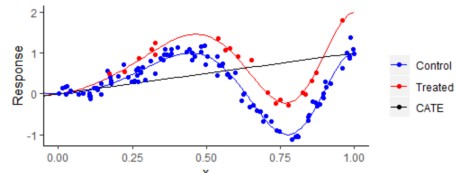

Figure 1: Illustrative toy example: the treatment effect can be much simpler than each PO function separately

often expected to be small relative to the complexity of the PO functions themselves [7], which, as in Fig. 1, could manifest in terms of CATE being a much simpler function than each PO.

One possible way of exploiting this is to estimate CATE directly using recently proposed model-agnostic *multi-stage* estimation strategies [7, 8, 18, 19]. However, these estimators output an estimate of CATE only and do not include an estimate of the untreated PO, which is often of independent interest in practical decision support problems – e.g. to trade off outcomes at baseline with the effectiveness of high-risk treatments [20]. From this literature, we borrow the insight that explicitly *targeting* learning strategies towards CATE, instead of only the POs, can lead to better estimators. However, instead of relying on multi-stage strategies, we operate within an *end-to-end* learning framework similar to [4] and recent extensions. Due to a focus on confounding, this line of work did not explicitly investigate how to exploit the similarity of the PO functions beyond them sharing a jointly learned feature space. Combining the two lines of work, we investigate how to use end-to-end approaches to output better estimates of both the POs *and* CATE by incorporating the assumption that the POs share much structure (which can result in a possibly simple CATE), as *inductive bias*.

We focus on a fundamental question that has received little explicit attention so far: ***How can we best exploit the structural similarities of the POs for CATE estimation?*** This question is crucial even in randomized experiments, making its solutions *orthogonal* to any of the sophisticated strategies developed to handle confounding. Therefore, we investigate approaches exploiting the shared structure of the POs which can be applied to *modify* existing modeling strategies, and thereby aim to provide guidance for *improving existing* CATE estimators along a new dimension. As such, the goal of this paper is not to promote the use of a specific approach, method or architecture. Rather, we aim to build systematic understanding and greater intuition of the (dis)advantages of different approaches. This is crucial in the context of CATE estimation, where model selection is notoriously difficult due to the absence of ground truth treatment effects in practice. We focus on gaining insight into the effect of different approaches relying on *the same* underlying ML method and use neural networks (NNs) due to their flexibility and popularity in related work, yet (variants of) the approaches we consider are applicable to many likelihood- or loss-based ML methods.

**Contributions** We investigate three approaches incorporating inductive biases for shared structure into the estimation of the POs: (1) a *soft approach*, which relies on regularization to encourage the PO functions to be similar and is hence easy to combine with existing methods, (2) a *hard approach*, which hardcodes an assumption on similarity into the model specification by reparametrization of the PO functions and (3) a *flexible approach*, in which we build on ideas from multi-task learning to design a new architecture for CATE estimation (FlexTENet), which adaptively *learns* what to share between the PO functions. We implement instantiations of all approaches using NNs and evaluate their performance across a wide range of semi-synthetic experiments, varying in the structural similarity of the PO functions. We empirically confirm that all approaches can improve upon baselines, including both end-to-end and multi-stage approaches, and present a number of insights into the relative strengths of each approach. We find that strategies significantly changing the model architecture – hard and flexible approaches – usually lead to the largest improvements, with FlexTENet performing best on average; yet even the simple soft approach often leads to notable performance increases – an insight that can easily be incorporated into any existing method with treatment-specific parameters.

## 2    Problem Definition and Key Challenges

Assume we observe a sample $\mathcal{D} = \{(Y_i, X_i, W_i)\}_{i=1}^n$, with $(Y_i, X_i, W_i) \overset{i.i.d.}{\sim} \mathbb{P}$. Here, $Y_i \in \mathcal{Y}$ is a continuous or binary outcome of interest, $X_i \in \mathcal{X} \subset \mathbb{R}^d$ a vector of possible confounders (i.e.

pre-treatment covariates) and $W_i \in \{0, 1\}$ is a binary treatment, assigned according to propensity score $\pi(x) = \mathbb{P}(W = 1|X = x)$. Using the Neyman-Rubin potential outcomes (PO) framework [9], our main interest lies in the individualized treatment effect: the difference between the PO $Y_i(1)$ if treatment is administered ($W_i = 1$) and $Y_i(0)$ if individual $i$ is not treated ($W_i = 0$). However, only one of the POs is observed as $Y_i = W_i Y_i(1) + (1 - W_i)Y_i(0)$. Therefore, we focus on estimating the conditional average treatment effect (CATE)

$$\tau(x) = \mathbb{E}[Y(1) - Y(0)|X = x] \tag{1}$$

which is the expected treatment effect for an individual with covariate values $X = x$. We operate under the standard identifying assumptions in the PO framework:

**Assumption 1.** *[Consistency, unconfoundedness and overlap] Consistency: If individual i is assigned treatment $w_i$, we observe the associated potential outcome $Y_i = Y_i(w_i)$. Unconfoundedness: there are no unobserved confounders, so that $Y(0), Y(1) \perp\!\!\!\perp W|X$. Overlap: treatment assignment is non-deterministic, i.e. $0 < \pi(x) < 1, \forall x \in \mathcal{X}$.*

### 2.1 The Key Challenges of CATE Estimation

As the ability to interpret a treatment effect estimate as causal ultimately relies on a set of *untestable* assumptions, the *unique* difficulty in making causal claims lies in using domain expertise to argue whether a treatment effect is identifiable [21]. Given identifiability, CATE estimation is a purely statistical problem – thus, if one is willing to rely on assumption 1, CATE can be estimated using observed data. A simple strategy for doing so (also known as the T-learner [7]) obtains regression estimates $\hat{\mu}_w(x)$ of $\mu_w(x) = \mathbb{E}[Y|X = x, W = w]$, applying standard supervised learning methods using only observed data for which $W = w$, and finally sets $\hat{\tau}(x) = \hat{\mu}_1(x) - \hat{\mu}_0(x)$. Yet, this seemingly straightforward solution is oblivious to two *statistical* challenges of CATE estimation:

**1. Confounding**: If $\pi(x)$ is not constant, then the distribution of covariates in treatment and control groups differs. Such imbalance can be the result of confounders, which are variables that affect both treatment selection and outcomes, and can be problematic when the PO functions are fit on the factual data using empirical risk minimization (ERM) because each problem is solved with respect to the wrong empirical distribution – namely $X \sim \mathbb{P}(\cdot|W = w)$ instead of $X \sim \mathbb{P}(\cdot)$. While this problem is not unique to CATE estimation – it is equivalent to the covariate shift problem encountered in e.g. domain adaptation [22] – it is usually emphasized as one of its main difficulties and motivated the literature on balanced representation learning [3, 4].

**2. CATE is the difference between two functions:** While supervised learning usually targets a single function, the goal of CATE estimation is to estimate *the difference* between two related functions most accurately – which may require different considerations than estimating each function separately. To see this, consider the MSE of estimating CATE and let $\epsilon_{sq}(\hat{f}(x)) = \mathbb{E}_{X \sim \mathbb{P}}[(\hat{f}(X) - f(X))^2]$ denote the MSE for an estimate $\hat{f}(x)$ of $f(x)$. If we simply were to estimate $\hat{\tau}(x) = \hat{\mu}_1(x) - \hat{\mu}_0(x)$ as the difference between two *separately* learned functions, we would have that (up to constants) $\epsilon_{sq}(\hat{\tau}(x)) \lesssim \epsilon_{sq}(\hat{\mu}_1(x)) + \epsilon_{sq}(\hat{\mu}_0(x)) \lesssim Rate_{\mu_1} + Rate_{\mu_0}$. The convergence rates $Rate_{\mu_w}$ depend on the used estimator and assumptions on e.g. smoothness or sparsity of the PO functions; a well-known example would be [23]'s nonparametric minimax rate. If we had oracle access to both POs and could regress $Y(1) - Y(0)$ on $X$ directly, we would have $\epsilon_{sq}(\hat{\tau}(x)) \lesssim Rate_\tau$. The assumption that $\tau(x)$ is often much simpler than each $\mu_w(x)$ separately [7] translates into $Rate_\tau < \max_w Rate_{\mu_w}$, highlighting that targeting CATE directly could lead to faster convergence. Similarly, any shared structure across the PO regression tasks could also be exploited to improve upon the simple additive bound above. These observations motivated much of this paper, as they have largely been neglected in the literature on end-to-end learning for CATE. They have, however, been the motivation for some model-agnostic multi-stage learners which we discuss next.

## 3 Related Work

**Direct and indirect meta-learners for CATE** Recent literature has developed a number of *model-agnostic* learning strategies for CATE estimation (also known as 'meta-learners' [7]). Within this class, we consider an *indirect* estimator any strategy that uses observed data to obtain regression estimates of the PO functions $\hat{\mu}_w(x)$ and then sets $\hat{\tau}(x) = \hat{\mu}_1(x) - \hat{\mu}_0(x)$. This includes [7]'s model-agnostic S- and T-learner; additionally the majority of model-specific ML-based CATE estimators also follow

an indirect strategy (including all NN-based estimators we discuss below). Conversely, we consider learners *direct* estimators if they target $\tau(x)$ directly. As $Y(1) - Y(0)$ is unobserved, multiple existing strategies construct *pseudo-outcomes* $Y_\eta$ for which it holds that $\mathbb{E}[Y_\eta|X = x] = \tau(x)$ for some nuisance parameters $\eta(x)$ which can be estimated from observational data. Different learners require estimation of different nuisance parameters, which often include propensity score $\pi(x)$ and/or PO functions $\mu_w(x)$. All direct estimation strategies that we are aware of – X-learner [7], R-learner [8], DR-learner [18], PW-learner and RA-learner [19] – proceed in a two-stage manner: they first obtain plug-in nuisance parameter estimates $\hat{\eta}(x)$, and then estimate $\tau(x)$ by regressing $Y_{\hat{\eta}}$ on $X$. Under some conditions, these two-stage learners can attain the oracle rate $Rate_\tau$. We give a more detailed overview of all meta-learner strategies in appendix A.

**NN-based CATE estimators** Complementing model-agnostic strategies, many adaptations of specific ML methods for CATE estimation have been proposed recently. We build on NN-based estimation strategies due to their flexibility and popularity in related work[1]. Much work on CATE estimation using NNs has focused on handling confounding, most prominently by learning shared and balanced feature representations for the two PO functions [3, 4]. Formally, this strategy entails jointly learning a shared feature map, and two PO-specific regression heads (fit using only the data of the corresponding treatment group) each parametrized by a NN. The output heads are then used for indirect estimation of CATE. Without further regularization, this leads to the TARNet specification, while CFRNet introduces a regularization term which encourages the network to learn representations that are balanced, i.e. have indistinguishable distributions across treatment groups [4]. Recent extensions investigated incorporating weighting strategies as an additional remedy for confounding into this framework [11, 12, 13, 14], considered targeting towards *average* treatment effects [27, 28] or allowed for both shared *and* private feature spaces for the PO functions [19].

**Relationship to multi-task learning** The architectures proposed in [4] and extensions effectively take a multi-task learning (MTL) approach to PO estimation, relying on *hard parameter sharing* [29] in the first $d_r$ layers of the used network, and no sharing in the top $d_h$ layers. While sharing a feature space will lead to some shared behavior between the estimated PO functions, it does not allow to fully exploit underlying similarity – e.g. if there are purely prognostic effects it might be better to share *some* information also between top layers. Below, we therefore investigate strategies to incorporate (additional) inductive bias for shared behavior into PO estimation. These are inspired by work in transfer learning [30], domain adaptation [31, 32] and, in particular, MTL [33, 34, 35], all allowing for flexible modeling of shared and task-specific aspects of a problem. Note, however, that MTL and CATE estimation have distinct statistical target parameters and goals – MTL is concerned with achieving a good *average performance in prediction of outcomes* across tasks, while the main target of CATE estimation is *estimating the expected difference between outcomes* – making it non-obvious a priori whether methods successful in the former will perform well in the latter problem.

# 4 Inductive Biases for CATE Estimation

In this section, we consider end-to-end approaches for incorporating the prior belief that $\mu_0(x)$ and $\mu_1(x)$ will share much structure (which can imply that $\tau(x)$ is simpler than $\mu_w(x)$) as inductive bias. Here, we define inductive bias as the mechanism by which some hypothesis functions are preferred over others during learning [36]. Throughout and for the remainder of this paper, we focus on exploiting the expected similarity between the PO functions, and disregard the impact of confounding on ERM – with the understanding that existing strategies, such as balancing or weighting, are orthogonal solutions that could readily be applied to complement any strategy we discuss.

## 4.1 Implicit inductive biases in indirect learners

We begin by examining the nature of the inductive biases present in popular indirect learners: a T-learner based on vanilla NNs (TNet) and TARNet. Recall that TARNet jointly learns a representation $\Phi : \mathcal{X} \to \mathcal{S}$, parametrized by a dense NN with $d_r$ layers and $n_r$ hidden units, and regression heads $h_w : \mathcal{S} \to \mathcal{Y}$, each parametrized by a dense NN with $d_h$ layers and $n_h$ hidden units. A TNet can be

---

[1]As we discuss in Appendix A, other popular estimators, which we do not consider further as we are interested in the effect of different approaches relying on *the same* underlying ML method, are based on Generalized Random Forests [24], Bayesian Additive Regression Trees [1, 25], Gaussian Processes [26, 6] and GANs [5].

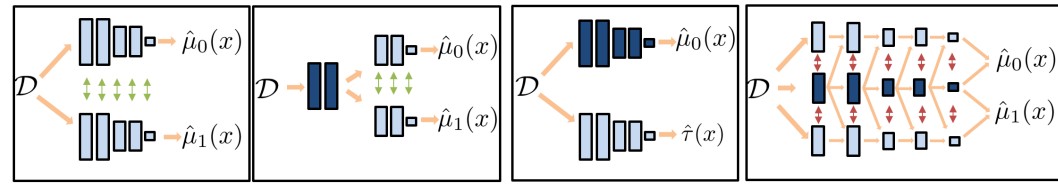

|  (1) Regularization for TNet (left) and TARNet (right) | (2) Reparametrization | (3) FlexTENet |

Figure 2: The three approaches under investigation. Dark layers indicate parameters shared between POs, light layers indicate private parameters. Green arrows indicate regularization encouraging parameters to be similar, red arrows indicate regularization that encourages orthogonalization.

seen as a special case of TARNet with $\Phi(x) = x$, i.e. no joint learning of feature spaces[2]. Below, we will refer to the weights in $h_w$ and $\Phi$ as $\Theta_{h_w}$ and $\Theta_\Phi$, respectively. During training, both TARNet and TNet use a loss function that takes the general form

$$\mathcal{L}_F + \lambda_1 \sum_{w \in \{0,1\}} \mathcal{R}(\Theta_{h_w}) \tag{2}$$

where $\mathcal{L}_F = \sum_{i=1}^n l(y_i, h_{w_i}(\Phi(x_i)))$, $\mathcal{R}$ is a regularizer (usually L2), which regularizes the complexity of each $h_w$ separately, and we dropped the regularization term for the shared representation – $\lambda_1 \mathcal{R}(\Theta_\Phi)$ – from the equation as it is not of relevance for the following discussions.

While the PO functions can thus share a jointly learned feature space, they are *not* encouraged to be similar beyond this. Instead, they are *separately* regularized to be a simple function, leading to their difference – $\tau(x)$ – being highly instable and hence to an implicit inductive bias that encourages treatment effect heterogeneity a priori (see also the discussions of this phenomenon in [8, 25]). This is neither in line with a scientific null hypothesis of no treatment effect (heterogeneity) nor the assumption that $\mu_1(x)$ and $\mu_0(x)$ should be close due to the existence of prognostic effects. The instability of $\hat{\tau}(x)$ can also be seen as a consequence of indirect learners not being well *targeted* towards CATE. In fact, using this regularization scheme, it is not even possible to control the complexity of CATE directly.

## 4.2 Explicit inductive biases for CATE estimation

We investigate three approaches modifying existing end-to-end learners to encourage shared structure in the POs and hence incorporate an inductive bias for simpler $\tau(x)$. Ordered by ease of implementation, we distinguish between (1) a soft approach relying on regularization, (2) a hard approach based on reparametrization and (3) a flexible approach (FlexTENet) which explicitly learns what to share between the POs. The architectures for each approach are depicted in Figure 2. Note that although all approaches are targeted at estimating CATE, only (2) directly outputs an estimate of $\tau(x)$.

**Soft approach – Regularization**  The most straightforward strategy to fixing the regularization-induced inductive bias towards heterogeneity discussed above would be to simply change how the PO functions are regularized. Instead of regularizing them separately, one could regularize the difference between the weights in the output heads – which ultimately determines $\tau(x)$ –, corresponding to an inductive bias towards small treatment effect heterogeneity[3]. Analogously to (2), this leads to a loss

$$\mathcal{L}_F + \lambda_1 \mathcal{R}(\Theta_{h_0}) + \lambda_2 \mathcal{R}(\Theta_{h_1} - \Theta_{h_0}) \tag{3}$$

Choosing $\lambda_2 > \lambda_1$ additionally reinforces the inductive bias towards simple $\tau(x)$. We further discuss how to set hyperparameters such as $\lambda_2$ (which is shared by all considered approaches) in Appendix B.1. This regularization-based approach is attractive because it is extremely easy to implement, is directly applicable to any loss-based method with treatment-group-specific parameters, has intuitive appeal and does not heavily constrain the functions the hypotheses are able to represent. At the same time, the latter point is a downside of this approach, since this might also result in only marginal gains.

---

[2]In our implementations, to give similar capacity to the resulting PO estimators and in analogy with a 'no parameter sharing' strategy, we give each $h_w$ in a TNet additional access to $d_r$ layers with $n_r$ hidden units.

[3]Note that, by convention, we regularize only the weights of NNs, and *not* biases (offsets), resulting in penalization only of non-constant treatment effects.

**Hard approach – Reparametrization** Instead of regularizing the difference between the PO functions, we could also *reparametrize* our estimators, a strategy that e.g. [37]'s LASSO for CATE estimation and [25]'s Bayesian Causal Forest rely on. Instead of estimating $\mu_0(x)$ and $\mu_1(x)$ separately, we can build on the identity $\mu_1(x) = \mu_0(x) + \tau(x)$ and effectively estimate $\tau(x)$ directly as an offset from $h_0(x)$, parametrized by a NN $h_\tau(x)$ with weights $\Theta_{h_\tau}$. This leads to a loss

$$\mathcal{L}_F + \lambda_1 \mathcal{R}(\Theta_{h_0}) + \lambda_2 \mathcal{R}(\Theta_{h_\tau}) \tag{4}$$

with $\mathcal{L}_F = \sum_{i=1}^n l(y_i, h_0(x_i) + w_i h_\tau(x_i))$ for continuous $y$, which seems analogous to (3) but parametrizes $\tau(x)$ explicitly – giving the investigator control over the complexity of $\tau(x)$ *directly* – and *hard-codes* the assumption that the shared structure between the POs is *additive*. This approach may be at a disadvantage if $\mu_1(x)$ is *simpler* than $\mu_0(x)$, since one would then be better off using the reverse parametrization $\mu_0(x) = \mu_1(x) - \tau(x)$. More generally, it is also possible that the relationship between $\mu_0(x)$ and $\mu_1(x)$ is not additive, e.g. if $\mu_1(x)$ is a simple transformation of $\mu_0(x)$ – say a multiplicative or logarithmic transformation (as is the case in the popular IHDP benchmark [1]) – such that a different parametrization would lead to an easier learning problem.

**Flexible approach – FlexTENet** Unfortunately, the parametrization leading to the easiest learning problem is usually not known in practice. Alternatively, one could thus rely on a strategy that can automatically and flexibly *learn* which information to (hard-)share between the PO functions. Inspired by architectures in MTL and domain adaptation that *explicitly* anticipate shared and private structure [31, 32, 35], we therefore propose a new architecture for treatment effect estimation, FlexTENet (Flexible Treatment Effect Network), as a final strategy. As depicted in Fig. 2, it has private ($\mu_w(x)$-specific) subspaces – which ultimately determine $\tau(x)$ – and a shared subspace in each layer (including the output layer), allowing the model to automatically learn which information to share *at each layer* of the network. In principle, any MTL method could be adapted for PO estimation, yet, as we discuss in appendix B.2, we propose the FlexTENet specification because it *generalizes* many existing strategies. Given its flexibility, we expect that such a general architecture should perform well *on average* – an appealing feature given that model-selection is nontrivial in CATE problems.

We implement FlexTENet using a specification matching TARNet to allow for direct comparisons, and consider $d_r + d_h$ layers, within which each private and shared subspace has $n_{k,p}$ and $n_{k,s}$ hidden units, respectively, where we let $n_{k,p} = n_{k,s} = \frac{1}{2} n_k$, $k \in \{r, h\}$, for simplicity. For layer $l > 1$, let $m_p^{l-1}$ and $m_s^{l-1}$ denote the output dimensions of shared and private subspace of the previous layer, let $\theta_s^l \in \mathbb{R}^{m_s^{l-1} \times m_s^l}$ denote the weights in the shared subspace, while $\theta_{p_w}^l \in \mathbb{R}^{(m_s^{l-1} + m_p^{l-1}) \times m_p^l}$ denotes the weights in each private subspace[4]. To discourage redundancy and encourage identification of private structure, we apply regularizers to orthogonalize the shared and private subspaces. Like [32, 35] we rely on [38]'s orthogonal regularizer $\mathcal{R}_o(\Theta_s, \Theta_{p_0}, \Theta_{p_1}) = \sum_{w \in \{0,1\}} \sum_{l=1}^L \|\theta_s^{l\top} \theta_{p_w, 1:m_s^{l-1}}^l\|_F^2$ where $\|\cdot\|_F^2$ denotes the squared Frobenius norm. This leads to the following loss function

$$\mathcal{L}_F + \lambda_1 \mathcal{R}(\Theta_s) + \lambda_2 \sum_{w \in \{0,1\}} \mathcal{R}(\Theta_{p_w}) + \lambda_o \mathcal{R}_o(\Theta_s, \Theta_{p_0}, \Theta_{p_1}) \tag{5}$$

where setting $\lambda_2 > \lambda_1$ adds inductive bias encouraging the shared space to be used first.

### 4.2.1 Underlying assumptions and theory

**Which assumptions do these approaches encode?** Motivated by real-world applications in which prognostic effects are often assumed stronger than predictive ones [7, 25], our central assumption is '*there is much shared structure between $\mu_0(x)$ and $\mu_1(x)$*'. This assumption is purposefully abstract, allowing it to manifest as different specific assumptions depending on the used ML method[5] and regularizer ($\mathcal{R}$); e.g. in regression with L0-penalty, $\tau(x)$ would be assumed linear and sparse, while in our case, using NNs with L2-penalty, the $\mu_w(x)$ are implicitly assumed to be close in some function class, with smooth differences. We thus consider inductive biases for $\tau(x)$ *relative* to the inductive bias in the original method used to estimate the $\mu_w(x)$; we effectively investigate how to best re-target

---

[4]The difference in input dimensions arises as we only allow communication from shared subspaces to private subspaces and not the reverse; refer to Appendix B.3 for pseudocode of a FlexTENet forward pass.

[5]While both hard and soft approach are directly applicable to any loss-based method, our flexible approach is specific to NNs. When using a different ML method, similar strategies could be constructed by adapting (i) existing MTL approaches or (ii) a model-agnostic approach similar to [31], creating PO-specific feature spaces.

these biases to control $\tau(x)$ explicitly. Further, there is a conceptual difference between assumptions in soft & flexible and hard approach; only the latter assumes shared structure to be additive.

**Why is shared structure inductive bias reasonable in this context?** Shared structure is a reasonable assumption in many practical applications as usually one would expect at least some similarities between treated and untreated individuals: intuitively speaking, receiving a drug will most likely not change all biological processes related to a disease progression in a patient and attending a job training program is unlikely to neutralize all characteristics determining an individual's salary. In medicine, for example, this has led to the well-known distinction between prognostic and predictive (effect-modifying) biomarkers [16, 17]; in our context, the strength of such prognostic information would determine the degree of shared structure. Further, assuming shared structure is compatible with explicit assumptions made in recent theoretical work on CATE meta-learners where CATE is assumed a simpler function than each of the POs [7, 18, 19] – which implies shared structure (i.e. a shared baseline function) between the POs. Additionally, a related (but much stronger) assumption is made in papers considering the popular semi-parametric 'partially linear regression model' analyzed in e.g. [39, 40]; here *all* nonparametric ('complex') structure is shared between POs, while the treatment effect is assumed parametric (often constant).

**Are there any theoretical expectations for CATE estimation performance?** All theoretical results focusing on estimating the difference between two functions that we are aware of originate in the literature on (i) CATE estimation [7, 18, 19] or (ii) transfer learning via offset estimation [41, 42]. They provide risk bounds for CATE estimation of the form $Rate_\tau + Rate_{Remainder}$, indicating that, if some remainder terms decay sufficiently fast, oracle rates for estimation of CATE can be attained. These results rely on two-stage estimation and stability conditions on the estimators, and as such are not directly applicable to our setting. Nonetheless, we hypothesize that our end-to-end learning strategies can match the performance of such estimators particularly in small sample regimes due to sharing of information between tasks (estimation of POs). It would therefore be an interesting next step to adapt theoretical results from MTL [43, 44] to analyze end-to-end strategies for CATE estimation, yet we consider this non-trivial as (i) MTL is concerned with the average performance over tasks, and not the performance on estimating task differences, and (ii) the number of tasks in our context is small ($T = 2$). We therefore defer theoretical analysis of our approaches to future research and focus on experimentally evaluating their performance below.

## 5 Experiments

### 5.1 Experimental setup

**Simulation settings**[6] As ground truth treatment effects are unobserved in practice, we use semi-synthetic setups based on real covariates and simulated $\mu_w(x)$. To systematically gain insight into the relative strengths of different strategies, we consider a number of setups (A-D) comprising a total of 101 simulation settings. We provide brief descriptions below; refer to Appendix C for more detail. For setups A&B, we use the ACIC2016 covariates ($n = 4802, d = 55$) of [45] but design our own response surfaces, allowing us to introduce our own 'experimental knobs' to enable structured evaluation of different approaches. We simulate response surfaces similar to [17] as

$$Y_i = c + \sum_{j=1}^d \beta_j(1 - W_i\omega_j)X_j + \sum_{j=1}^d \sum_{l=1}^d \beta_{j,l}(1 - W_i\omega_{j,l})X_jX_l + W_i \sum_{j=1}^d \gamma_jX_j + \epsilon_i \quad (6)$$

where $\beta_j, \gamma_j, \omega_j, \beta_{j,l}, \omega_{j,l} \in \{0, 1\}$ are sampled as Bernoulli random variables, and the probability of non-zero $\beta_j, \beta_{j,l}$ is fixed throughout. In Setup A, we set all $\omega_j = \omega_{j,l} = 0$ and control the complexity of $\tau(x) = \sum \gamma_jX_j$ by varying the expected proportion $\rho$ of non-zero $\gamma_j$; thus $\mu_0(x)$ is sparser than $\mu_1(x)$. Conversely, in setup B we set all $\gamma_j = 0$ and instead induce treatment effect heterogeneity by sampling non-zero $\omega_j, \omega_{j,l}$ with probability $\rho$. These cancel some prognostic effects for the treated, such that the complexity of $\tau(x)$ increases as the complexity of $\mu_1(x)$ decreases; here $\mu_1(x)$ is thus simpler than $\mu_0(x)$. In both settings, knob $\rho$ determines the complexity of $\tau(x)$ through the number of predictive features; as $\rho$ increases, $\mu_1(x)$ becomes less similar to fixed $\mu_0(x)$. We randomly assign $n_0 \in \{200, 500, 2000\}$ to control and $\{n_1 \in \{200, 500, 2000\} : n_1 \leq n_0\}$ to treatment, creating different levels of sample imbalance (*without* confounding), and use 500 units for testing.

For setups C&D, we use the IHDP benchmark ($n = 747, d = 25$), into which [1] introduced confounding, imbalance (18% treated) and incomplete overlap. We use [1]'s original simulation as

---

[6]Code to replicate all experiments is available at `https://github.com/AliciaCurth/CATENets`

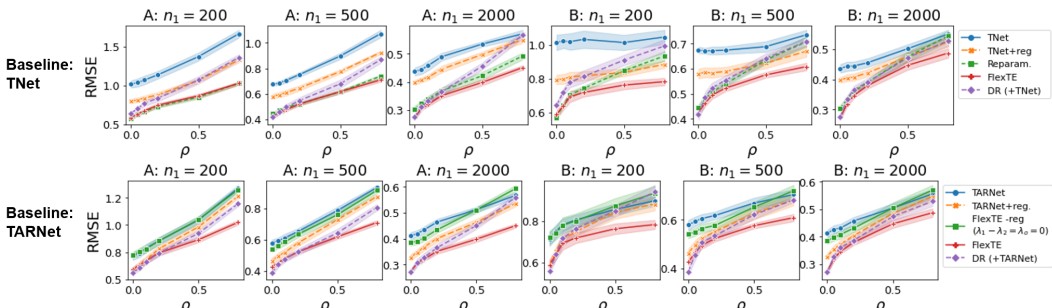

Figure 3: RMSE of CATE estimation by $\rho$ for multiple $n_1$ for setup A and B, using TNet (top) and TARNet (bottom) as baseline, at $n_0 = 2000$. In each graph, as $\rho$ increases $\tau(x)$ becomes more complex because $\mu_1(x)$ becomes less (more) sparse. Avg. across 10 runs, one standard error shaded.

setup C, consisting of simulated expected POs $\mu_0^C(x) = \exp((X + 0.5)\beta)$ and $\mu_1^C = c + X\beta$, where $\beta$ is a randomly sampled sparse vector. Both response surfaces share the dependency on $X\beta$ albeit in different functional form; here the assumptions of additive similarity and $\tau(x)$ being simpler than $\mu_w(x)$ do *not* hold and there is very strong heterogeneity. We add setup D, in which these assumptions do hold, by slightly altering the response surfaces to $\mu_0^D(x) = \mu_0^C(x)$ and $\mu_1^D(x) = \mu_0^C(x) + \mu_1^D(x)$ (i.e. an additive treatment effect). Here, we use the 90/10 train-test splits of [4]'s IHDP-100 benchmark to evaluate in- and out-of-sample performance. In Appendix E, we report additional results (102 additional settings) using the original ACIC2016 response surfaces and the real-world Twins dataset.

**Models** We compare the three approaches to TNet and TARNet as baseline indirect estimators and [18]'s DR-learner, a direct two-stage estimator, which we implement using both TNet and TARNet in the first stage. We also considered RA- [19], R- [8] and X-learner [7], but found them to perform worse than the DR-learner on average (see appendix D). We perform further baseline comparisons, using DragonNet [27] and SNet [19] in appendix D. To ensure fair comparison across all models and all experiments, we fix hyperparameters across all models within each experimental study and ensure that each estimator/output function has access to the same number of hidden units across all models[7] and effectively use each model "off-the-shelf". We discuss implementation further in Appendix C.

### 5.2 Results and Discussion

Below, we discuss our findings. We begin with setups A&B and evaluate performance on CATE estimation with imbalanced data (where $n_0 = 2000$ unless stated otherwise). We perform ablations to gain insight into the relative effect of the different components of each strategy, consider the effect of smaller $n_0$ and consider performance on PO estimation. Finally, we consider setups C&D and ACIC2016 & Twins (Appendix E) to consolidate our findings on well-established benchmark datasets. We highlight some aspects of the results here, and present additional findings in appendix D (including varying $n_0$, further results on estimating the POs, analysing the weights of a trained FlexTENet and additional baseline comparisons).

**Comparison to indirect learners:** *All three approaches improve upon baseline indirect learners in CATE estimation with large $n_0$ (setups A&B); regularization brings smallest gains, reparametrization works well in setup A but not B, and FlexTENet performs best on average.* Considering Fig. 3, we make a number of interesting observations. First, all three strategies consistently improve upon the respective baseline in almost all settings and largest improvements are made relative to TNet with $n_1$ small. Additionally, the performance of the three approaches is closest when $\rho$ is small ($\tau(x)$ is sparse). Second, gains for regularization are most apparent with TNet for $n_1 \leq 500$. Adding regularization to TARNet brings relatively smaller improvements because its baseline error is smaller. Third, for $n_1 \leq 500$, FlexTENet and reparametrization perform equivalently in setups A but not in B, which is to be expected because the latter uses an impractical parametrization for B, while FlexTENet can freely adapt to the underlying problem structure. Fourth, comparing FlexTENet and TARNet, we observe that when we use the FlexTENet architecture without additional regularisation ($\lambda_o = 0$ and $\lambda_1 = \lambda_2$), their performance is similar, while incorporating additional inductive bias, encouraging identification of predictive and prognostic effects, leads to substantial improvement of FlexTENet over TARNet across all setups. Finally, FlexTENet performs best on average and seems

---

[7]Due to different degrees of parameter sharing, a TNet has more parameters than TARNet and FlexTENet.

to have a particular advantage for larger $n_1$ and $\rho$, where the latter indicates that FlexTENet is not only well-equipped to handle prognostic effects, but also treatment effect heterogeneity.

**Comparison to multi-stage learners:** *Regularization rarely outperforms the DR-learner while FlexTENet and reparametrization generally do.* Unlike the other two approaches, regularization rarely outperforms the DR-learner, yet it often matches its performance (particularly in setup B). Both FlexTENet and reparametrization outperform the DR-learner, yet performance of reparametrization and DR-learner seem to converge as $n_1$ increases, particularly in setup B. In Appendix D, we additionally investigate using the soft- and flexible approach as the first stage of DR- and X-learner.

**Ablation study:** *All components contribute to better performance.* We study ablations for setup A, $n_1 = 200$ (Fig. 4) to gain insight into the relative effect of different components of each approach. We find that setting $\lambda_1 < \lambda_2$, i.e. regularizing CATE more heavily than the POs, indeed led to additional (albeit smaller) improvement. For FlexTENet, orthogonal regularization alone adds more than setting $\lambda_1 < \lambda_2$, yet together they lead to the greatest improvement.

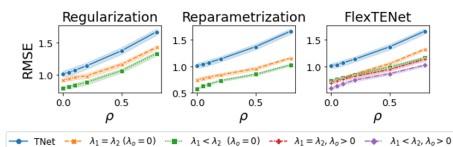

Figure 4: RMSE of CATE estimation by $\rho$ for ablations ($n_1 = 200$, setup A). Avg. across 10 runs, one standard error shaded.

**Effect of $n_0$:** *Improvements over baseline are less substantial for smaller $n_0$.* We investigate the effect of having a smaller set of control units $n_0$ in Fig. 5. While all strategies continue to outperform TNet, we observe that their gains are much smaller than for $n_0 = 2000$ (Fig 3). Additionally, we observe that FlexTENet appears to perform somewhat less well for small $\rho$, indicating that it may need more training data than other approaches due to its flexibility.

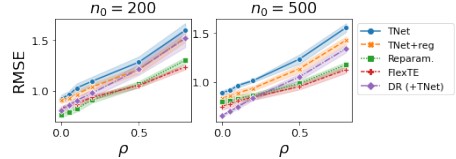

Figure 5: RMSE of CATE estimation by $\rho$ for $n_0 = 200, 500$ ($n_1 = 200$, setup A). Avg. across 10 runs, one standard error shaded.

**PO estimation:** *Most approaches also lead to improvements in PO estimation.* We consider whether the three approaches improved not only estimation of CATE but also of the POs separately in Fig. 6. For estimation of $\mu_1(x)$ at $n_1 = 500$, we observe that almost all strategies improve upon TNet, but make two interesting observations: First, the performance gap between TNet and its regularized version is qualitatively smaller than the performance gap in estimating CATE (particularly in setup B),

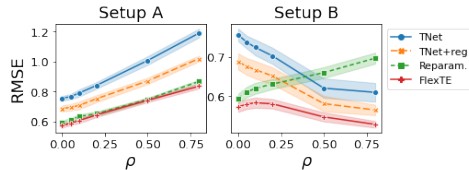

Figure 6: RMSE of $\mu_1(x)$ estimation by $\rho$ for $n_1 = 500$ for setup A and B. Averaged across 10 runs, one standard error shaded.

indicating that regularization could have a larger impact on improving the CATE estimate than the PO estimate. Second, the reparametrization approach is indeed unable to handle setup B where $\mu_1(x)$ becomes progressively simpler. Interestingly, even though it performs *worse* at estimating $\mu_1(x)$ than TNet for large $\rho$, Fig. 3 still showed better performance at estimating $\tau(x)$ in this case – indicating that the reparametrization approach is better targeted towards CATE estimation than PO estimation.

**Further benchmark results:** *Performance on IHDP, ACIC2016 and Twins setups reinforces findings and confirms expectations.* Even though the IHDP dataset is smaller, subject to confounding and has limited overlap, performance across setups C and D (Table 1) largely confirms the findings previously discussed. The only major difference is induced by setup C in which $\tau(x)$ is *not* simpler than each $\mu_w(x)$ separately and an additive parametrization does not lead to the easiest learning problem; as

Table 1: Normalized[8] in- & out-of-sample RMSE of CATE, setup C & D. Avg. across 100 runs, standard error in parentheses.

|  | C, in | C, out | D, in | D, out |
|---|---|---|---|---|
| TNet | 0.32 (.01) | 0.34 (.01) | 0.29 (.01) | 0.29 (.01) |
| TNet + reg | 0.30 (.01) | 0.32 (.01) | 0.26 (.01) | 0.26 (.01) |
| DR (+TNet) | 0.35 (.01) | 0.37 (.01) | 0.22 (.01) | 0.22 (.01) |
| TARNet | 0.29 (.01) | 0.31 (.01) | 0.22 (.01) | 0.23 (.01) |
| TARNet + reg | 0.28 (.01) | 0.31 (.01) | **0.20** (.01) | **0.20** (.01) |
| DR (+TARNet) | 0.33 (.01) | 0.34 (.01) | **0.20** (.01) | **0.20** (.01) |
| Reparam. | 0.39 (.01) | 0.40 (.01) | **0.20** (.01) | **0.20** (.01) |
| FlexTENet | **0.27** (.01) | **0.29** (.01) | 0.22 (.01) | 0.23 (.01) |

---

[8]Due to the exponential in $\mu_0(x)$, RMSE varies by orders of magnitude across runs and seems unsuitable to assess relative performance. We report RMSE normalized by standard deviation of observed outcomes (see appendix for details and unnormalized results).

expected both reparametrization and the DR-learner perform poorly in this scenario (yet methods using shared representations are nonetheless able to exploit the shared dependence of the POs on $X\beta$). TARNet in combination with the soft approach and FlexTENet show the best average performance across the two setups, where the weaker performance of FlexTENet on setup D is in line with previous findings for small $n_0$ and simple $\tau(x)$. The results on the ACIC2016 simulations and the real-world Twins dataset presented in appendix E further confirm the relative performance of the different methods we observed throughout.

## 6    Conclusion

We found that altering the inductive biases in end-to-end CATE estimators can lead to performance increases and match, or improve upon, the finite sample performance of multi-stage learners targeting CATE directly. We observed that all our approaches were particularly useful when only one treatment group had abundant samples, a situation arising naturally in practice when a new treatment is introduced. We found that strategies which change the model architecture – reparametrization and FlexTENet – led to the largest improvements: The former seems to be the best choice in smaller samples when treatment effects are additive and $\mu_0(x)$ is simpler than $\mu_1(x)$, while FlexTENet showed the best average performance due to its flexibility and handled high heterogeneity well, but required more training data when $\tau(x)$ is very simple. Additionally, we also found that the simple strategy of changing the regularizer leads to notable performance increases, an insight that could directly be incorporated into many existing methods with treatment-specific parameters.

Here, we limited our attention to a single ML method (NNs) to *isolate* the properties of different approaches. Therefore, an interesting next step would be to consolidate our findings by applying our approaches using different underlying methods. Finally, as all approaches rely on the ability to manipulate loss (or likelihood) functions, it would be interesting to consider how to equip methods that do not have easily manipulable loss functions (e.g. random forests) with similar inductive biases.

## Acknowledgments and Disclosure of Funding

We thank anonymous reviewers as well as members of the vanderschaar-lab for many insightful comment and suggestions. AC gratefully acknowledges funding from AstraZeneca.

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
