# On Inductive Biases for Heterogeneous Treatment Effect Estimation
# Appendix

**Alicia Curth**
University of Cambridge
amc253@cam.ac.uk

**Mihaela van der Schaar**
University of Cambridge
University of California, Los Angeles
The Alan Turing Institute
mv472@cam.ac.uk

This appendix is organized as follows: We first present an additional literature review (Section A) and then discuss additional details of the proposed approaches (Section B).We then provide further experimental details (Section C), discuss additional results on Setups A-D (Section D) and finally present results on additional benchmark datasets (Section E). We include the NeurIPS checklist in Section F.

## A  Additional Literature Review

Here, we present a detailed overview of existing model-agnostic "meta-learner" strategies for CATE estimation which can be implemented using *any* ML method, a notion originally introduced in [1] and expanded on in [2, 3, 4]. As in the main text, we distinguish between indirect and direct estimators for CATE, and will finally briefly discuss ML-based strategies that do *not* fall in the meta-learner class because they rely on a specific ML-method.

**Indirect Estimators** The S- and T-learner discussed in [1] are two model-agnostic learning strategies that estimate CATE *indirectly*. The S-learner fits a **s**ingle regression model $\hat{\mu}(x, w)$ by concatenating the covariate vector $X$ and the treatment indicator $W$ into $X'$ and then regressing $Y$ on $X'$, providing a final CATE estimate indirectly as $\hat{\tau}(x) = \hat{\mu}(x, 1) - \hat{\mu}(x, 0)$. The T-learner fits **t**wo regression models (one $\hat{\mu}_w(x) = \mathbb{E}[Y|W = w, X = x]$ for each treatment group) *separately* using only observations for which $W = w$, and provides a final CATE estimate as $\hat{\tau}(x) = \hat{\mu}_1(x) - \hat{\mu}_0(x)$.

**Multi-stage Direct Estimators** A number of meta-learners have been proposed recently which target CATE *directly* through a multi-stage estimation procedure. We will first discuss four learning strategies that rely on *pseudo-outcome* regression, and will then discuss [2]'s R-learner, which uses a loss-based approach.

We follow the exposition in [4] distinguishing between three classes of pseudo-outcome regression-based meta-learners, which use a first stage to obtain estimates $\hat{\eta}$ of (a subset of) the nuisance parameters $\eta = (\mu_0(x), \mu_1(x), \pi(x))$. Pseudo-outcome regression then proceeds by obtaining an estimate of $\hat{\tau}(x)$ by regressing a pseudo-outcome $\tilde{Y}_{\hat{\eta}}$ (based on nuisance estimates $\hat{\eta}$) on $X$ directly. For all considered pseudo-outcomes it holds that $\mathbb{E}_{\mathbb{P}}[\tilde{Y}_\eta|X = x] = \tau(x)$ – they are unbiased for CATE when $\eta$ is known. Inspired by the well-known estimation strategies for the *average* treatment effect (ATE), there are three straightforward strategies for doing so:

(1) Regression Adjustment (RA): The RA-learner [4] uses a regression-adjusted pseudo-outcome

$$\tilde{Y}_{RA,\hat{\eta}} = W(Y - \hat{\mu}_0(X)) + (1 - W)(\hat{\mu}_1(X) - Y) \tag{1}$$

in the second stage. The X-learner proposed in [1] is a variant of this estimator: Instead of performing one regression in the second stage, they perform two separate regressions for each term in the sum,

leading to two CATE estimators $\hat{\tau}_1(x)$ and $\hat{\tau}_0(x)$ that are then combined into a final estimate using $\hat{\tau}(x) = g(x)\hat{\tau}_1(x) + (1 - g(x))\hat{\tau}_0(x)$ for some weighting function $g(x)$.

(2) Propensity Weighting (PW): The PW-learner uses a pseudo-outcome based on the Horvitz-Thompson transformation [5]

$$\tilde{Y}_{PW,\hat{\eta}} = \left( \frac{W}{\hat{\pi}(X)} - \frac{1-W}{1-\hat{\pi}(X)} \right) Y \tag{2}$$

(3) Doubly Robust (DR): [3]'s DR-learner has pseudo-outcome

$$\tilde{Y}_{DR,\hat{\eta}} = \left( \frac{W}{\hat{\pi}(X)} - \frac{(1-W)}{1-\hat{\pi}(X)} \right) Y + \left[ \left( 1 - \frac{W}{\hat{\pi}(X)} \right) \hat{\mu}_1(x) - \left( 1 - \frac{1-W}{1-\hat{\pi}(X)} \right) \hat{\mu}_0(X) \right] \tag{3}$$

which is based on the doubly-robust AIPW estimator [6] and is unbiased if either propensity score *or* outcome regressions are correctly specified.

For two-stage estimators targeting the treatment effect directly using pseudo-outcomes, it can be shown that $\epsilon_{sq}(\hat{\tau}(x)) \leq \epsilon_{sq}(\hat{\tau}_\eta(x)) + R^2_{\hat{\eta}}(x)$ if appropriate sample splitting is used and the used estimator fulfills some stability condition [3]. Here, $\epsilon_{sq}(\hat{\tau}_\eta(x))$ converges at the oracle rate, and $R^2_{\hat{\eta}}$ is a learner-specific remainder term. Two-stage learners thus converge at oracle rates if the remainder term decays sufficiently fast, which is faster than indirect learners if $\tau(x)$ is simple. Due to the double robustness property, the remainder term of the DR-learner always converges faster than the other two learners, and it is in general unlikely that RA-learner can attain the oracle rate. For a comprehensive overview of convergence of the different pseudo-outcome-based meta-learners, refer to [4].

Finally, [2] also propose a two-stage algorithm that estimates CATE directly but is based on a loss-based strategy instead of pseudo-outcome regression. The R-learner is based on [7] approach for semiparametric regression, and uses orthogonalization with respect to the nuisance functions $\pi(x)$ and $\mu(x) = \mathbb{E}[Y|X = x]$ (the unconditional outcome expectation). Their first stage obtains estimates $\hat{\pi}(x)$ and $\hat{\mu}(x)$, which are then used in a second stage estimating $\tau(x)$ directly based on the following loss:

$$\arg\min_\tau \sum_{i=1}^n \left[ \{Y_i - \hat{\mu}(X_i)\} - \{W_i - \hat{\pi}(X_i)\}\tau(X_i) \right]^2 + \mathcal{R}(\tau(\cdot)) \tag{4}$$

Like the DR-learner, this learner is doubly robust due to orthogonalization with respect to both outcome and propensity score estimate, and also arises from the perspective of 'orthogonal statistical learning' [8].

For theoretical guarantees to hold for any of the two-stage learners, first and second stage have to be performed on separate folds of the data, either by splitting the data or by using cross-fitting [9]. In our experiments, we do use the full data for both stages as we found that sample splitting/cross-fitting deteriorates performance, most likely due to the small sample sizes.

**Model-specific CATE estimators** Many methods proposed in related work do not fall within the meta-learner class because they rely on the properties of a *specific* ML method. Some methods rely on a strategy that can be seen as a hybrid between [1]'s S- and T-learner, sharing *some* information between regression tasks but not all; this includes most multi-task approaches ([10] and extensions, but also e.g. [11, 12]). GANITE [13], another popular strategy, relies on GANs to learn counterfactual distributions instead of conditional expected values. Additionally, as discussed in the main text, [14]'s LASSO for CATE estimation and [15]'s Bayesian Causal Forest rely a the reparametrization strategy (while [16]'s popular BART-based CATE estimator is a simple S-learner). Finally, [17]'s Causal Forest relies on a local moment equation inspired by the Robinson transformation, and could hence be seen as a forest-based version of the R-learner.

# B    Additional discussion of proposed approaches

## B.1    Hyperparameter tuning

It is a well-known problem in the CATE estimation literature that model selection is nontrivial due to the absence of counterfactuals in practice. The testable implications of the shared structure bias, as encoded by hyperparameters such as $\lambda_2$, however, are different for (i) the PO estimation and (ii)

the CATE estimation problems, which is a feature that we would suggest to exploit in choosing hyperparameter settings. That is, while the usefulness of the inductive biases for estimation *of CATE* cannot easily be verified, their usefulness for estimating *the POs* can be verified through cross-validation on held-out factual observations.

Unfortunately, good performance on estimation of the POs is not sufficient. To see this, note that the tuples of PO estimates $(\mu_0(x) + e(x), \mu_1(x) + e(x))$ and $(\mu_0(x) + e(x), \mu_1(x) - e(x))$ [where $e(x)$ is an error and $\mu_w(x)$ is the ground truth] will in expectation give exactly the same MSE when evaluated using held-out factual observations. Yet, the first tuple will make no error in estimating CATE, while the second tuple makes an error of $2e(x)$. This highlights that error can either compound or cancel across the POs, therefore making it possible that a hyperparameter setting resulting in better fit on the POs also results in worse fit on CATE, and vice versa. Because an estimators' performance on estimating CATE remains unobservable in practice, a good inductive bias is needed to choose between models that have equal predictive performance on the POs. As we discuss in the paper, we consider preferring shared structure (or a simple CATE) a reasonable choice for this in many practical applications.

**Suggested approach for setting hyperparameters.**  As a simple heuristic to set hyperparameters such as $\lambda_2$ in practice, we would thus recommend the following scheme that trades off between imposing an assumption ("CATE is most likely simple") and factual performance:

1. Start with $\lambda_2$ small, and keep increasing it while held-out predictive performance does not decrease.
2. Set $\lambda_2$ to its largest value for which predictive performance did not deteriorate.

In a sufficiently flexible (overparameterized) model class in which multiple PO estimators induce the same empirical performance, such a scheme allows to pick the hyperparameter setting resulting in the least complex CATE while remaining compatible with factual observations.

**Illustrative results**  In Fig. 1 we present illustrative results on Setup B with $n_0 = n_1 = 2000$, and observe that following our heuristic of increasing $\lambda_2$ until factual performance deteriorates would almost always lead to choosing the best hyperparameter setting; for both hard and flexible approach this suggests a switch from $\lambda = 10^{-1}$ to $\lambda_2 = 10^{-2}$ as $\rho$ increases[1]. Further, we note that performance in terms of factual prediction is indeed often much closer than CATE estimation performance [2].

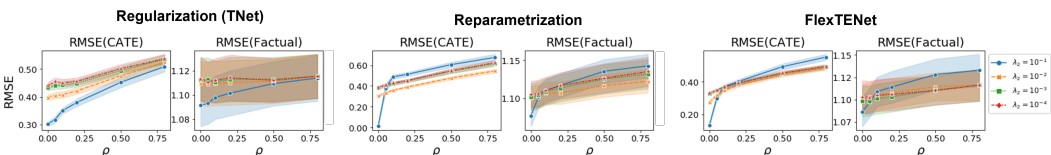

Figure 1: Effect of $\lambda_2$ on RMSE of estimating CATE and factual RMSE (test set), for regularization (left), reparametrization (middle) and flexible (right) approach, by $\rho$ in Setup B at $n_0 = n_1 = 2000$. Avg. across 10 runs, one standard error shaded.

## B.2   FlexTENet as a generalization of existing architectures

As summarized in Table 1, existing architectures for CATE estimation arise as special cases of FlexTENet (all of which heavily restrict the flow of information within the network): As we reduce the width of all shared layers while increasing the width of private layers, FlexTENet approaches a TNet. Conversely, if we increase the width of the bottom shared layers and do the reverse for the top

---

[1]Note that, as we discuss in section C.2, we fixed all hyperparameters throughout all experiments as tuning would have been computationally prohibitive given the amount of experimental settings and models considered. Throughout, we used $\lambda_1 = 10^{-4}$ and $\lambda_2 = 100\lambda_1 = 10^{-2}$ to induce a substantial difference between the two.

[2]Note also that for factual evaluation observations have random normal noise with $\sigma = 1$, which explains why $RMSE(factual) > 1$ while $RMSE(CATE) < 1$.

Table 1: Existing architectures for CATE estimation arise as special cases of FlexTENet

| Method | $(n_{r,s}, n_{r,0}, n_{r,1})$ | $(n_{h,s}, n_{h,0}, n_{h,1})$ | Communicat. subspaces |
|---|---|---|---|
| FlexTENet | $(n_{r,s}, n_{r,0}, n_{r,1})$ | $(n_{h,s}, n_{h,0}, n_{h,1})$ | Yes |
| TARNet | $(n_r, 0, 0)$ | $(0, n_h, n_h)$ | No |
| TNet | $(0, n_r, n_r)$ | $(0, n_h, n_h)$ | No |
| SNet | $(n_{r,s}, n_{r,0}, n_{r,1})$ | $(0, n_h, n_h)$ | No |
| Reparam. | $(n_r, 0, n_r)$ | $(n_h, 0, n_h)$ | No |

layers, FlexTENet becomes TARNet. Relative to TARNet, this architecture thus not only incorporates inductive bias towards shared behavior in the output heads, but also explicitly anticipates the existence of purely predictive features by allowing for both shared and PO-specific features instead of enforcing one joint representation (which might erroneously discard features that are relevant only for one of the POs). Therefore, FlexTENet also generalizes the SNet class discussed in [4], which includes PO-specific feature spaces[3]. Finally, FlexTENet without private subspaces for $\mu_0(x)$ and without communication between shared and private subspaces is equivalent to the reparametrization approach. Relative to existing strategies, we expect that such a general architecture should perform well *on average*, given that it not only encompasses them, but also allows for more general forms of shared structure which other architectures cannot exploit.

## B.3   Pseudocode of a FlexTENet forward pass

---
**Algorithm 1:** FlexTENet forward pass.

---
**Input :** Testing data X
         Trained FlexTENet `flex`
**for** $i \leftarrow 1{:}flex.n\_layers$ **do**
   **if** $i{==}1$ **then**
      `x_shared = flex.shared_layers[i](X)`
      `x_po0 = flex.po0_layers[i](X)`
      `x_po1 = flex.po1_layers[i](X)`
   **end**
   **else**
      `x_po0 = flex.po0_layers[i](Concatenate(x_shared,x_po0))`
      `x_po1 = flex.po1_layers[i](Concatenate(x_shared,x_po1))`
      `x_shared = flex.shared_layers[i](x_shared)`
   **end**
**end**
**if** $flex.binary\_y$ **then**
   `y0_hat = Sigmoid(x_shared+x_po0)`
   `y1_hat = Sigmoid(x_shared+x_po1)`
**end**
**else**
   `y0_hat = x_shared+x_po0`
   `y1_hat = x_shared+x_po1`
**end**
**return** $y0\_hat, y1\_hat$

---

---
[3]The general SNet specification in [4] also includes propensity estimators as additional output heads, which could be added to FlexTENet if needed.

## C  Experimental details[4]

### C.1  Simulation details

We considered setups A-D in the main text, as they allowed us to evaluate the performance of the different approaches along axes that are most relevant to the problem we are trying to solve. Unlike much related work (e.g. [10, 18, 19]) we therefore did not focus on varying the level of confounding, but instead considered (i) the level of alignment between the PO functions and (ii) differences in data availability between the two treatment groups. While (i) directly determines how applicable/useful the underlying inductive biases are, (ii) is of interest because we wanted to test whether a large control group allows to distill prognostic effects better, such that less treatment data is needed to determine predictive effects.

Major differences across setups include that:

- in setups A, B, and D, the treatment effects are additive, while setup C inherits a non-additive treatment effect from [16]'s IHDP simulation
- in setups A & B there is no confounding as treatment assignment is random, while in C & D, there is confounding and incomplete overlap
- in setups A & B, the response surface includes only polynomial terms of the covariates (linear, squares and first-order interactions) while in C & D the baseline outcome exponentiates a linear predictor
- in setups A & D, $\mu_0(x)$ is simpler than $\mu_1(x)$, while the reverse is true in setups B & C.

Note that, while Setups C & D corresponded to one simulation setting each, setups A & B as presented in Fig. 3 in the main text corresponded to 33 simulation settings (as $\rho$ and $n_1$ vary). By further varying $n_0$, we consider another 66 settings in section D.3. Below, we give further insight into the data and data-generating processes (DGPs) we used.

### C.1.1  Setups A and B

We use the data from the Collaborative Perinatal Project provided[5] in the first Atlantic Causal Inference Competition (ACIC2016) [20] for our first set of experiments. The original dataset has $d = 58$ covariates, of which we exclude the 3 categorical ones. Of the remaining 55 covariates, 5 are binary, 27 are count data and 23 are continuous. We process all covariates according to the transformations used in the competition[6], which transforms count into binary covariates and standardizes continuous variables. We use the transformed data for the simulations and as input to all models.

As discussed in the main text, we simulate response surfaces in setup A according to

$$Y_i = c + \sum_{j=1}^{d} \beta_j X_j + \sum_{j=1}^{d}\sum_{l=1}^{d} \beta_{j,l} X_j X_l + W_i \sum_{j=1}^{d} \gamma_j X_j + \epsilon_i \tag{5}$$

where $\epsilon_i \sim N(0,1)$, $\beta_j \sim \mathcal{B}(0.6)$ and $\gamma_j \sim \mathcal{B}(\rho)$. We include squared terms of all continuous covariates and additionally include each variable randomly into one interaction term, for both of which we then simulate coefficient $\beta_{j,l} \sim \mathcal{B}(0.3)$. We chose for each coefficient to be binary to avoid large variances in the scale of POs and CATE across different runs of a simulation, such that RMSE remains comparable across runs.

For setup B, we instead use $\gamma_j = 0$ and simulate

$$Y_i = c + \sum_{j=1}^{d} \beta_j(1 - W_i\omega_j)X_j + \sum_{j=1}^{d}\sum_{l=1}^{d} \beta_{j,l}(1 - W_i\omega_{j,l})X_j X_l + \epsilon_i \tag{6}$$

where only $\omega_j, \omega_{j,l} \sim \mathcal{B}(\rho)$ differ from above. While in setup A non-zero $\gamma_j$ induce treatment effect heterogeneity as $\mu_1(x)$ has *more* terms than $\mu_0(x)$, in setup B non-zero $\omega_j$ induce treatment effect heterogeneity, giving $\mu_1(x)$ *less* terms than $\mu_0(x)$.

---

[4]Code to replicate all experiments is available at `https://github.com/AliciaCurth/CATENets`

[5]We retrieve the data from `https://jenniferhill7.wixsite.com/acic-2016/competition`

[6]We use the code available at `https://github.com/vdorie/aciccomp/blob/master/2016/R/transformInput.R`

### C.1.2 Setups C and D (IHDP)

For setups C & D, we build on the Infant Health and Development Program (IHDP) benchmark used in [10] and extensions, created by [16]. The underlying dataset belongs to a real randomized experiment targeting premature infants with low birth weight with an intervention, containing 25 covariates (6 continuous, 19 binary) capturing aspects related to children and their mothers. The benchmark dataset was created by excluding a non-random proportion of treated individuals (those with nonwhite mothers). The final dataset consists of 747 observations (139 treated, 608 control), and overlap is not satisfied (as $\pi(x)$ is not necessarily non-zero for all observations in the control group). While the covariate data is real, the outcomes in our setup "C" are simulated according to setup "B" described in [16], which satisfies $Y(0) \sim \mathcal{N}(exp((X + W)\beta), 1)$ and $Y(1) \sim \mathcal{N}(X\beta - \omega, 1)$ with $W$ an offset matrix, $\omega$ is set such that the average treatment effect on the treated is equal to 4, and the coefficient $\beta$ has entries in $(0, 0.1, 0.2, 0.3, 0.4)$, where each entry is independently sampled with probabilities $(0.6, 0.1, 0.1, 0.1, 0.1)$. We use the 100 repetitions of the simulation provided by [10][7]. For our setup "D" we change only the response surface of the treated to $Y(1) \sim \mathcal{N}(exp((X + W)\beta) + X\beta - \omega, 1)$.

### C.2 Implementation details

In our implementations, we use components similar to those used in [10] for all networks. In particular, we use dense layers with exponential linear units (ELU) as nonlinear activation functions. We train with Adam [21], minibatches of size 100, and use early stopping based on a 30% validation split. As stated in the main text, we fixed equivalent hyperparameters across all methods within any experiments to not conflate hyperparameter tuning with the value of the different strategies. We set $n_r = 200$ and $n_h = 100$ throughout, and use $d_r = 1$ and $d_h = 1$ for setups A & B and $d_r = 2$ and $d_h = 2$ for setups C & D (excluding one additional dense output layer) for all estimators – i.e. for TARNet, but also for each other function (e.g. the second stage of a two-step learner is parameterized by $d_r$ and $d_h$ layers of $n_r$ and $n_h$ units each). Note that TNet and TARNet with similarity regularization share the identical architecture with their 'vanilla' counterparts, and differ only in the regularization term and in that we initialise the $\Theta_{h_w}$ weights with the same random initialisation for both heads. For FlexTENet we set $n_{k,p} = n_{k,s} = \frac{1}{2}n_k$ for $k \in \{r, h\}$. Throughout, we use $\lambda_1 = 0.0001$, $\lambda_2 = 100\lambda_1$ (to induce a substantial difference) and $\lambda_o = 0.1$, where we chose the magnitude of $\lambda_o$ by testing it on toy data. Further, we use all two-step learners *without* sample splitting or cross-fitting which we found to deteriorate performance, particularly in the smaller sample sizes.

All models were implemented in our own python codebase, using jax [22]. All experiments were conducted using Python 3.8 on Ubuntu 20.04 OS with a Intel i7-8550U CPU with 4 cores. Creating the results in Figure 3 and Table 1 (main text) took about 12h each.

## D   Additional results (setups A-D)

Below, we present additional results. First, we consider additional baselines: we compare the performance of different two-stage learners across setups A & B (D.1) and consider further indirect learners as baselines (DragonNet and SNet, D.2). Then, we consider the effect of $n_0$ in setups A & B (D.3), present additional results on PO estimation (D.4), and then move to analyzing the learned weights of a FlexTENet (D.5). We also consider the effect of using our approaches as first-stage (nuisance) estimators for two-step learners (D.6). Finally, we discuss the necessity of scaling the results in setups C & D (D.7)

### D.1   Comparison of two-stage learners across setups A and B

When comparing the performance of DR-learner, R-learner, RA-learner and X-learner (Fig 2), we observe that the DR-learner shows best performance *on average* – which is why we used it as a baseline in the main text. Nonetheless, the R-learner can outperform it for small $\rho$, while the X-learner can outperform it for large $\rho$.

---

[7]Available at `https://www.fredjo.com/`

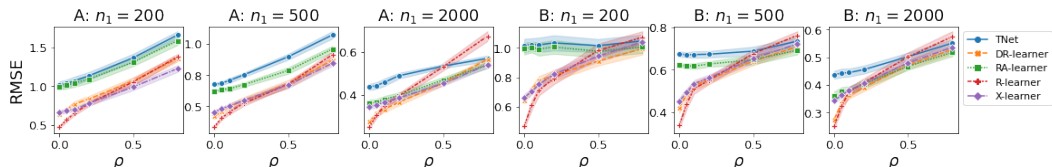

Figure 2: RMSE of CATE estimation for two-step learners using TNet as baseline by $\rho$ for different $n_1$ with $n_0 = 2000$ for setup A ($\mu_0(x)$ is simpler) and B ($\mu_1(x)$ is simpler). Recall that, in each graph, as $\rho$ increases $\tau(x)$ becomes more complex because $\mu_1(x)$ becomes less (more) sparse. Avg. across 10 runs, one standard error shaded.

## D.2 Experiments with additional indirect learners

In Fig. 3, we present additional results using DragonNet [23] and SNet [4] on setups A & B. As there is no confounding in these setups, the propensity head of DragonNet does not contribute to performance, such that TARNet and DragonNet perform virtually equivalently across all settings, and the soft approach improves also the performance of DragonNet.

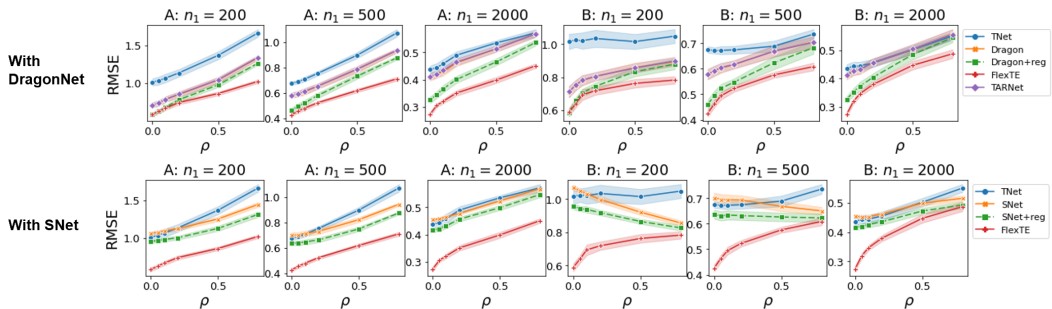

Figure 3: RMSE of CATE estimation using DragonNet (top) and SNet (bottom) as additional baselines, by $\rho$ for multiple $n_1$ at $n_0 = 2000$, for setup A and B. Avg. across 10 runs, one standard error shaded.

For SNet, to facilitate comparison, we consider a variant of the original formulation in [4] without propensity head, such that SNet reduces to having only 3 feature spaces - a shared and two PO-specific feature spaces just like FlexTENet. As such, SNet and FlexTENet differ mainly in their output heads. Further, we implement SNet using the same orthogonal regularizer as FlexTENet to allow for fair comparison (this differs from the original orthogonal regularizer used in [4], which induces more sparsity as it relied on a l1-norm). We observe that the soft approach improves also the performance of SNet, and that FlexTENet substantially outperforms SNet despite their similarities; highlighting that hard-sharing in the output heads is useful.

Results for Setups C and D are presented in table 2. We observe that DragonNet once more performs very similar to TARNet; here it does perform slightly better, possibly due to the presence of confounding in this dataset. However, as $\mu_w(x)$ and $\pi(x)$ are not well-aligned, the gains are marginal. SNet performs poorly overall, which is to be expected as there are no

Table 2: Normalized in- & out-of-sample RMSE of CATE estimation for additional benchmarks and selected methods, setup C & D. Avg. across 100 runs, standard error in parentheses.

|  | C, in | C, out | D, in | D, out |
|---|---|---|---|---|
| TNet | 0.320 (.008) | 0.337 (.008) | 0.290 (.007) | 0.290 (.008) |
| TNet + reg | 0.301 (.008) | 0.324 (.008) | 0.260 (.005) | 0.262 (.006) |
| TARNet | 0.294 (.008) | 0.315 (.008) | 0.225 (.007) | 0.226 (.007) |
| TARNet + reg | 0.285 (.008) | 0.306 (.008) | 0.205 (.006) | 0.205 (.006) |
| FlexTENet | **0.268** (.009) | **0.293** (.009) | 0.224 (.005) | 0.230 (.006) |
| DragonNet | 0.289 (0.008) | 0.310 (0.008) | 0.222 (0.006) | 0.223 (0.007) |
| DragonNet + reg | 0.282 (0.008) | 0.304 (0.008) | **0.203** (0.006) | **0.203** (0.006) |
| SNet | 0.327 (0.014) | 0.356 (0.013) | 0.261 (0.008) | 0.266 (0.009) |
| SNet + reg | 0.316 (0.013) | 0.345 (0.012) | 0.252 (0.008) | 0.258 (0.009) |

$\mu_w(x)$-specific features in these setups. Further, both baselines benefit from the addition of the soft approach.

### D.3 The effect of $n_0$ in setups A and B

In Fig. 4 we investigate the effect of having a smaller number of control samples, $n_0 = 200, 500, 1000$ instead of $n_0 = 2000$ as in the main text, at $n_1 = 200$. We observe that in setup A, increasing $n_0$ leads to convergence of performance of both regularization and DR-learner as well as of FlexTENet and reparametrization. Interestingly, in setup B, we observe divergence of the different methods.

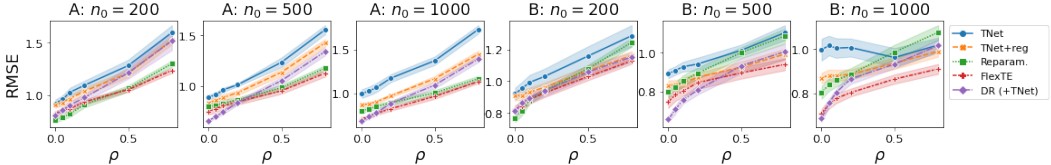

Figure 4: RMSE of CATE estimation by $\rho$ for $n_0 = 200, 500, 1000$ for setup A and B with $n_1 = 200$ using TNet as baseline. Avg. across 10 runs, one standard error shaded.

Additionally, we consider increasing the number of observations in treatment and control group equally, i.e. $n_0 = n_1$, for $n_w \in \{200, 500, 1000\}$ in Fig. 5 ($n_0 = n_1 = 2000$ is included in the results in the main text). As briefly discussed in the main text, the gain of using each approach is much smaller in the balanced than in the imbalanced setting, but the conclusions regarding the relative performance of each approach remain largely the same. Most salient is the strong performance of the DR-learner for small $\rho$ (often outperforming all other methods) and in setup B throughout (often matching the performance of FlexTENet for large $\rho$).

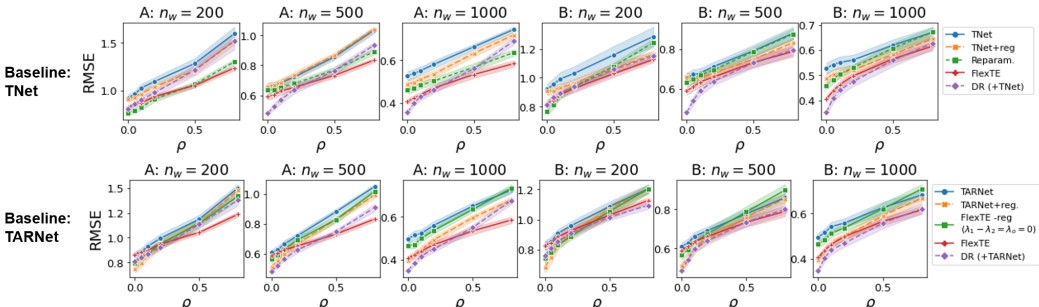

Figure 5: RMSE of CATE estimation by $\rho$ for $n_0 = n_1 = 200, 500, 1000$ for setup A and B, using TNet (top row) and TARNet (bottom row) as baseline. Avg. across 10 runs, one standard error shaded.

### D.4 Additional results for PO estimation

In Fig. 6, we observe that the impact of all approaches on $\mu_0(x)$ estimation is negligible for small $n_1$ (at $n_0 = 2000$), but that there are some improvements for $n_1 = 2000$. Further, the inability of reparametrization to handle setup B is even more apparent for $n_1 = 2000$.

### D.5 Analysis of FlexTENet weights

In Fig. 7 we analyze what the FlexTENet learns by considering the average L2-norm of the weights of each hidden unit for each layer and subspace. We observe that in the lower layers, most weight is on the shared component for all $\rho$, and that the weight on the $\mu_1(x)$-specific ($\mu_0(x)$-specific) component increases with increasing $\rho$ in setup A (B), as expected. Further, while in the output layer the weight on the shared layer remains approximately constant with $\rho$ in setup A, it decreases in setup B where the shared component becomes sparser as $\rho$ increases.

### D.6 Using the the approaches as improved first-stage estimators for two-stage learners

While meta-learners are usually implemented using vanilla first-stage estimators (i.e. in our case a separate neural network for each nuisance estimation task), we investigate here whether our improved

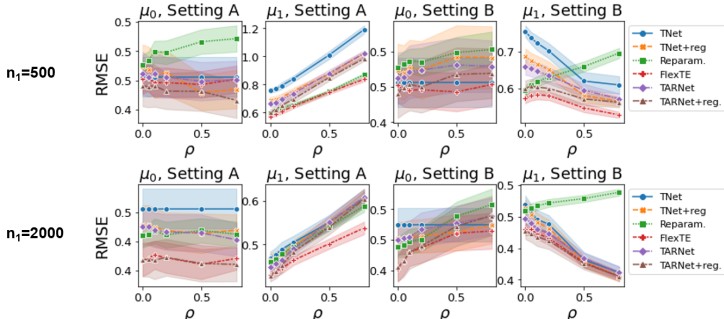

Figure 6: RMSE of $\mu_w(x)$ estimation by $\rho$ for $n_1 = 500$ (top) and $n_1 = 2000$ (bottom) for setup A and B at $n_0 = 2000$. Avg. across 10 runs, one standard error shaded.

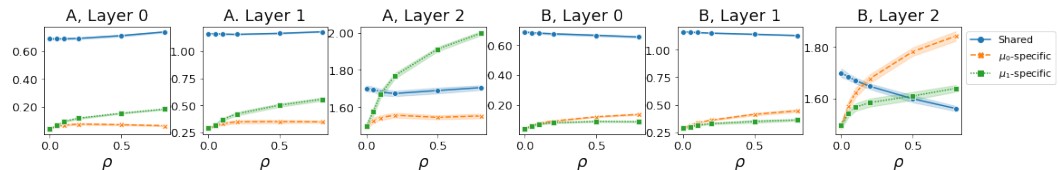

Figure 7: Average L2-norm of the weights of each hidden unit for each layer and subspace of the FlexTENet by $\rho$, for setup A (left) and B (right) at $n_0 = n_1 = 2000$. Avg. across 10 runs, one standard error shaded.

indirect estimators can be used to improve performance of two-stage learners, here X- and DR-learner. In Fig. 8 for the X-learner and Fig. 9 for the DR-learner, we find that, while there are some performance increases, improvements in the second stage are much smaller than improvements to the first stage and, for the case of FlexTENet, the second stage does not improve upon the first-stage estimate. As discussed in section C.2, we did not use sample-splitting in our experiments as we found this to deteriorate performance for all meta-learners (especially in smaller sample sizes); this might be one reason for this finding.

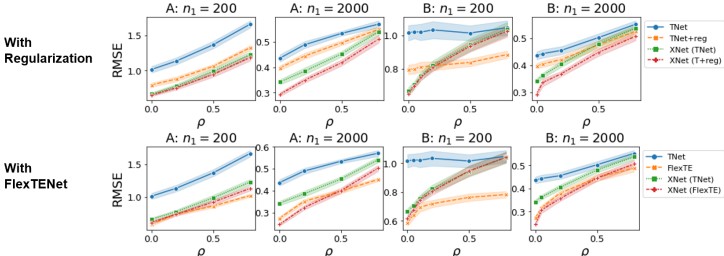

Figure 8: RMSE of CATE estimation for the X-learner using different methods as first-stage estimators, by $\rho$ for different $n_1$ with $n_0 = 2000$ for setup A & B. Avg. across 10 runs, one standard error shaded.

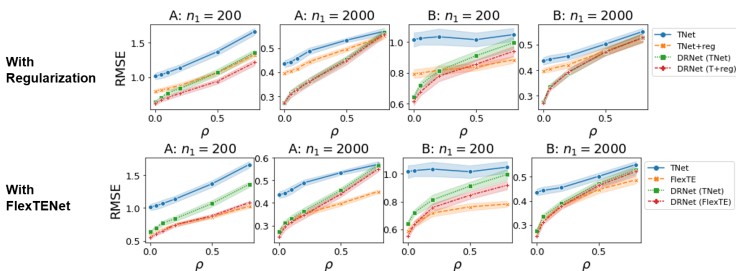

Figure 9: RMSE of CATE estimation for the DR-learner using different methods as first-stage estimators, by $\rho$ for different $n_1$ with $n_0 = 2000$ for setup A & B. Avg. across 10 runs, one standard error shaded.

## D.7 Scaling of IHDP results (setups C & D)

In setups C & D, we observed that the scale of RMSE of CATE estimation varied by orders of magnitude across different runs of the simulation due to the exponential regression specification in the response surface, making performance in terms of RMSE incomparable across runs. We found that by averaging RMSE across runs, the relative performance was dominated by runs with high variance in factual outcomes (which arise in runs in which many variables enter the exponential specification). Therefore, we report RMSE normalized by standard deviation of the observed factual training data in the main text. As Figure 10 highlights for the example of TARNet, this leads to much more well-behaved distributions of RMSE in both setups.

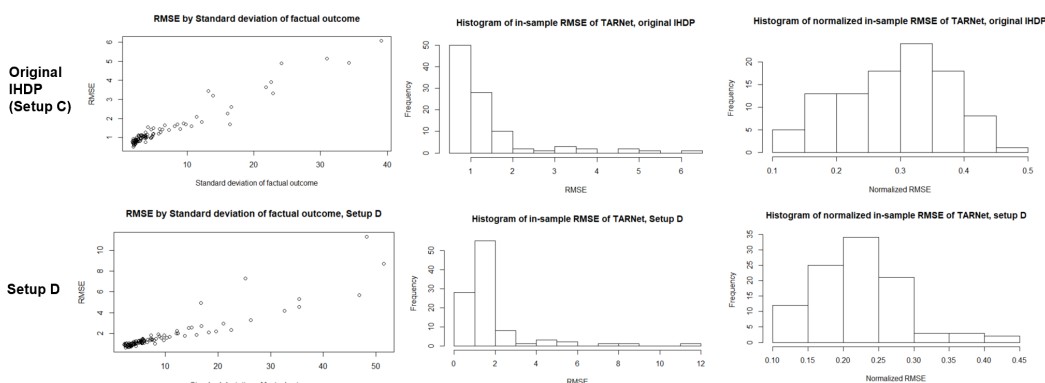

Figure 10: In-sample RMSE of TARNet by standard deviation of factual outcomes in training sample, histogram of in-sample RMSE across runs and histogram for normalized in-sample RMSE across runs for Setup C (top) and Setup D (bottom)

In Table 3 we report unnormalized results on the two IHDP setups for completeness. The results for TARNet differ from those reported in [10] for three main reasons: First, we used the IHDP-100 benchmark, and not the 1000 replications used in [10]. Second, as we highlighted above, the RMSE scores are clearly not normally distributed with the same mean such that some runs have much larger influence than others, and reported standard errors do not necessarily reflect the right confidence levels. Third, we used our own implementations and hyperparameter settings/architectural specification for these experiments, which differ slightly from those used in [10].

Table 3: Unnormalized in- and out-of-sample RMSE of CATE estimation on the two IHDP setups. Averaged across 100 runs, standard error in parentheses.

|  | C, in | C, out | D, in | D, out |
|---|---|---|---|---|
| TNet | 1.572 (0.134) | 1.775 (0.198) | 2.857 (0.438) | 2.854 (0.445) |
| TNet + reg | 1.418 (0.107) | 1.720 (0.193) | 2.355 (0.285) | 2.382 (0.293) |
| DR (+TNet) | 1.681 (0.134) | 1.924 (0.198) | 1.764 (0.171) | 1.740 (0.158) |
| TARNet | 1.384 (0.107) | 1.690 (0.196) | 1.772 (0.188) | 1.804 (0.204) |
| TARNet + reg | 1.350 (0.107) | 1.657 (0.197) | 1.623 (0.164) | 1.647 (0.181) |
| DR (+TARNet) | 1.574 (0.125) | 1.801 (0.183) | 1.463 (0.128) | 1.448 (0.121) |
| Reparam. | 2.136 (0.219) | 2.297 (0.269) | 1.394 (0.098) | 1.392 (0.092) |
| FlexTENet | 1.226 (0.099) | 1.536 (0.182) | 1.966 (0.239) | 2.057 (0.261) |

# E    Additional benchmark datasets

Below we present results on additional benchmark datasets: in section E.1 we consider the original response surfaces simulated for ACIC2016, and in section E.2 we consider performance on the Twins dataset with real outcomes. Throughout, we use the same hyperparameters as for setups A & B.

## E.1    Original ACIC2016

As an additional benchmark dataset, we consider performance on the original simulations of ACIC2016 [20]. The 77 settings vary in the complexity of response surfaces and the degrees of confounding, overlap and TE heterogeneity. They are based on the same covariates as simulations A and B, but differ in treatment assignment and response surfaces. In the competition, covariates were provided to participants without preprocessing, hence we start with the unprocessed dataset and, similar to [19], we standardize all covariates and drop the three categorical variables as none of the considered methods are well-suited to handle them. We report results on 20% test-sets for each of the 77 settings (averaging across 10 repetitions each).

In Fig. 11, we present results. Due to high variation in RMSE across the simulation runs (even within the same setting), we report RMSE(method)/RMSE(baseline) for TNet and TARNet as baseline. We note that our original findings on relative performance of the approaches from the main text hold up across a majority of settings. Apart from that, we found little meaningful insights into sources of performance variation across different settings, possibly due to the very high variation in simulated response surfaces within and across settings.

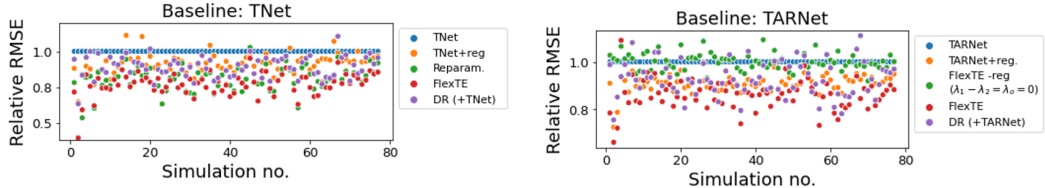

Figure 11: RMSE relative to TNet (left) and TARNet (right) on the 77 simulation settings from ACIC2016. Averaged across 10 runs each.

## E.2    Twins

Evaluation of treatment effect estimators on real data is usually prohibited by the absence of ground-truth treatment effects and counterfactuals in practice. Twin studies in which each twin is assigned a different treatment therefore present an interesting exception: under the assumption of equivalence between two twins, realisations of both potential outcomes are observed. The Twins dataset considered by [24, 13] contains one-year mortality outcomes for 11400 pairs of twins with 39 relevant covariates[8].

---

[8]We obtained the preprocessed dataset used in [13] from the authors, which is derived from the data provided by [25]

Here, the treatment is 'being heavier at birth', so that this data can be used to evaluate the effect of birthweight on infant mortality.

The outcome in this dataset is binary and (fortunately) imbalanced; mortality rates over the full data are $16.1\%$ and $17.7\%$ for treated and untreated, respectively. Due to the binary nature of the data and this imbalance, the signal for the presence of treatment effects (which necessitates observing opposite outcomes for pairs of twins), is relatively weak and noisy. Therefore, we use a large test set and hold out $50\%$ (5700 pairs of twins) for testing. For training, we randomly select one twin from each pair with (constant) probability $p_{treat} \in \{0.1, 0.25, 0.5, 0.75, 0.9\}$ to investigate the performance of each method on imbalanced data as in the main text, but now with *real* data. Additionally, we vary the number of training examples $n_{train} \in \{500, 1000, 2000, 4000, 5700\}$ to assess the sample efficiency of different methods; leading to 25 different settings considered for the Twins data.

**Metrics** As the true $\tau(x)$ and $\mu_w(x)$ are unobserved, we can only use realisations of $Y(w)$ to evaluate all models. First, as [13] we consider the RMSE on the observed counterfactual difference, i.e. $\sqrt{\frac{1}{n}\sum_{i=1}^{n}\left((y_i(1)-y_i(0))-(\hat\mu_1(x_i)-\hat\mu_0)\right)^2}$, as a metric to evaluate the quality of the treatment effect estimate. Because $Y$ is binary, $y_i(1)-y_i(0) \in \{-1,0,1\}$ while $\hat\mu_1(x_i)-\hat\mu_0$ is not, and this metric is very noisy. Therefore, we additionally consider $Y(1)-Y(0)$ as the target in a 3-class classification problem, where

$$\mathbb{P}(Y(1)-Y(0)=t|X=x) = \begin{cases} \mu_0(x) \times (1-\mu_1(x)) & \text{if } t=-1 \\ (1-\mu_0(x)) \times \mu_1(x) & \text{if } t=1 \\ \mu_0(x) \times \mu_1(x) + (1-\mu_0(x) \times (1-\mu_1(x)) & \text{if } t=0 \end{cases} \quad (7)$$

if we assume that the two potential outcomes are conditionally independent. We can compute $\hat{\mathbb{P}}(Y(1)-Y(0)=t|X=x)$ for all models which predict potential outcomes using (7), and evaluate its fit on the real data using standard classification metrics. Here, we report the area under the receiver-operating curve (henceforth: AUC). Further, similar to [24], we also evaluate predictive performance on each of the POs separately through the AUC.

### E.2.1 Performance on estimating CATE

In Fig. 12 and Fig. 13, we present results on performance of estimating the counterfactual difference, measured by RMSE and AUC, respectively. We observe that Reparametrization and FlexTENet perform almost equivalently and best throughout as measured by both metrics. Both are most robust to changes in $p_{treat}$, and, contrary to most other methods, perform close to optimal already with only 500 samples. Further, the regularization approach improves upon its baselines also in this setting. The conclusions on relative performance made in the main text thus largely hold up also here; the main difference is that here FlexTENet performs very well also for small $n_{train}$. This, and the near-equivalence of FlexTENet and reparametrization approach, might indicate that there is both significant shared structure *and* relevant (*additive*) heterogeneity between the (unknown) POs in the Twins dataset.

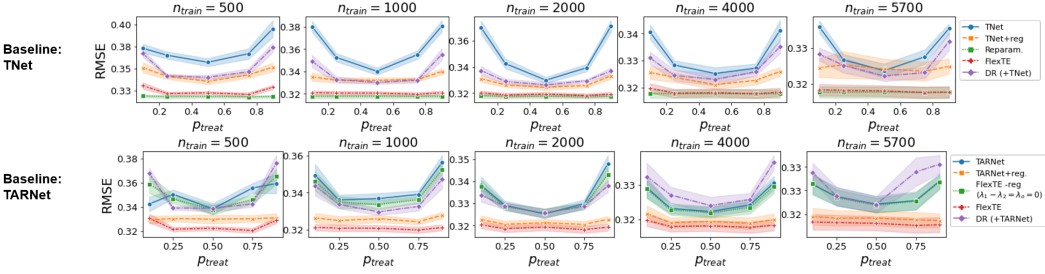

Figure 12: RMSE on the counterfactual difference (lower is better), by $p_{treat}$ for different $n_{train}$ using TNet (top) and TARNet (bottom) as baseline. Avg. across 10 runs, one standard error shaded.

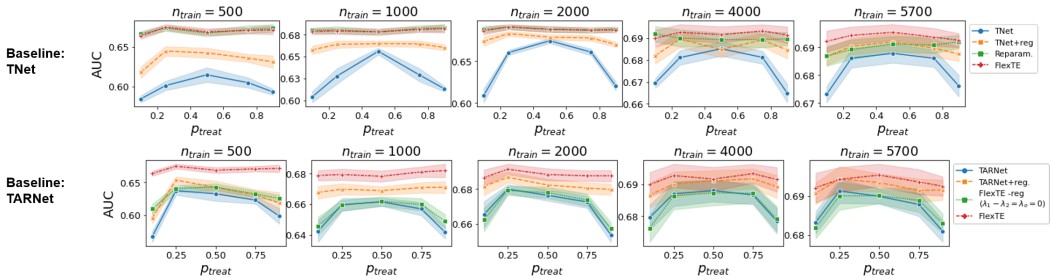

Figure 13: AUC on the counterfactual difference (higher is better), by $p_{treat}$ for different $n_{train}$ using TNet (top) and TARNet (bottom) as baseline. Avg. across 10 runs, one standard error shaded.

#### E.2.2 Performance on estimating the POs

Finally, we consider performance on estimating the POs separately in Fig. 14. In the left panels we plot AUC on each potential outcome for different levels of $p_{treat}$, and observe that all approaches can significantly improve the performance on estimating the POs when there is imbalanced treatment assignment; most likely this is because they provide additional supervision for the underrepresented treatment arm. In the right panels we plot AUC for underrepresented treatment arms by different levels of $n_{train}$, and observe that FlexTENet and the reparametrization approach provide such supervision most sample-efficiently: they reach near-optimal performance with a fraction of the available samples.

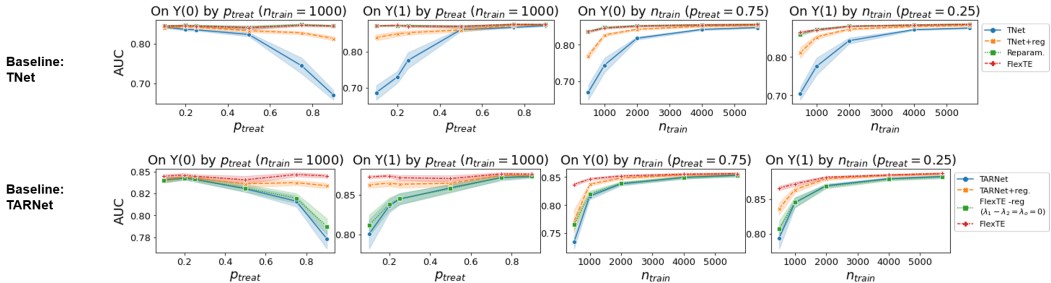

Figure 14: AUC on the potential outcomes (higher is better), by $p_{treat}$ (left) and by $n_{train}$ (right) using TNet (top) and TARNet (bottom) as baseline. Avg. across 10 runs, one standard error shaded.