# OpenReview forum: "On Inductive Biases for Heterogeneous Treatment Effect Estimation"
_NeurIPS.cc/2021/Conference — NeurIPS 2021 Spotlight_

### Official Review · Reviewer_9oJ5 · 2021-06-30

**Rating:** 8
**Confidence:** 3

**Summary:**

This paper contributes an investigation of the inductive biases of various neural network architectures in the context of conditional average treatment effect estimation. Specifically, the authors put forth the hypothesis that $\mu_0(x) = E[Y|X=x, W=0]$ and  $\mu_1(x) = E[Y|X=x, W=1]$ have similar structure and thus $\tau(x) = \mu_1(x) - \mu_0(x)$ *can* be simpler than either $\mu_w(x)$.

The authors investigate three approaches: (1) a two path (one per treatment) neural network, where the parameters $\theta_w$ of both paths are regularized to be point-wise close, (2) reparametrization whereby they explicitly learn a $\mu_0$ and a $\tau$ and set $\mu_1$ as $\mu_0 + \tau$, and (3) the authors' proposed approach (FlexTENet) where there are "private" functions $\mu_w$ for each treatment $w$ and a "shared" subspace in each layer.

**Limitations And Societal Impact:**

Yes.

**Main Review:**

Thanks to the authors on the hard work on this paper. It is obvious that they have put in time and thought to address the concerns of previous reviewers (ICML 2021).  The paper is ready for publication in my opinion, especially considering the wealth of additional experimental evidence in the appendix. CATE is hard, and having promising neural network architectures is important.

The work is largely clear and well-written. A few questions:

(1) The section that actually describes FlexTENet starts on line 229. I found it lacked the details I was expecting. Some pseudocode or math in the appendix would be helpful. It's not clear exactly what the arrows mean in Figure 2.3. When 2 orange arrows go into the same block, does that mean the inputs are concatenated? That is, the private $\theta$ are always being mixed with the shared $\theta$?

(2) A lot of this work is presented as being built on the idea that $\tau$ may be easier to estimate than $\mu_w$. However, I am unclear as to when this would actually be the case in practice. For example, if the true $\mu_w$ was a decision tree, then $\mu_1 - \mu_0$ would be a difference of two decision trees, which is more complex than either one. From what I can tell, your toy example in Figure 1 is a lot of the justification of this idea. Can you justify it further? Perhaps using data from the twin study? Also, just because the experimental results are promising, does that mean you've found evidence of the hypothesis that $\tau$ is simpler than $\mu_w$?

(3) In lines 401 and 402 you wrote "The former seems to be the best choice ... when ... $\mu_0$ is simpler than $\mu_1$." Is it possible to fit 2 variants of the reparametrization model, where $h_0$ estimates $\mu_0$ in one case and $h_1$ estimates $\mu_1$ in the other case, and then choose the better one as part of the hyperparameter optimization? Would this approach significantly improve the reparametrization model model? After all, one or the other will be more complex in practice, although they could be of very similar complexity. Does the reparametrization model fail when $\mu_0$ and $\mu_1$ are of comparable complexity?

**Time Spent Reviewing:**

3

---

> ### Author Response · Authors · 2021-08-10
> **Response to Reviewer 9oJ5**
>
> Thank you for your thoughtful comments and suggestions. We are very grateful for your appreciation of our work! Below, we give answers to the three groups of questions you raised.
>
> ***
> __*(1) Details on FlexTENet.*__
> Thank you for pointing out that FlexTENet could be discussed in further detail; as per your suggestion we will include pseudocode of a forward pass in the appendix, in addition to the descriptions and loss functions already presented in the main text.
>
> To answer your question, you are indeed correct in assuming that two arrows entering the same block in Fig. 2.3 indicates that the inputs are concatenated. This specifically means that each private subspace gets as input the concatenation of private and shared activations from the previous layer, while each shared subspace gets as input only the shared activations from the previous layer.
>
> __*(2) When would there be shared structure and a simple $\tau(x)$ in practice?*__
> The underlying specifications and complexities of $\mu_w(x)$ and $\tau(x)$ are indeed unknown in reality, and could differ substantially across different fields of application. Therefore, (implicit) assumptions on how data is generated in practice have formed across the literature. We observe that most of our references which simulate their own PO functions in experiments indeed assume that there is a shared baseline outcome (‘shared structure’) and an additive CATE in their DGPs, e.g. [2, 7, 8, 14, 17, 18, 19, 24]; this homogeneity is despite them originating in different fields (computer science,(bio)statistics and econometrics). Indeed, in all practical applications we are familiar with (mainly in medicine & economics), one would expect at least *some* similarities (‘shared structure’) between treated and untreated individuals: intuitively, receiving a drug will most likely not change *all* biological processes related to a disease progression in a patient, and attending a job training program is unlikely to neutralize *all* characteristics determining an individual’s salary. Generally speaking, the more such shared structure there is between an individual’s POs, the lower the relative complexity of $\tau(x)$.
>
> To link this back to your tree example, note that assuming a simple $\tau(x)$ (or shared structure between the $\mu_w(x)$) could mean that the underlying trees $\mu_w$ that generated the data share some common splits close to the root node, and that treatment treatment effect heterogeneity is induced only through differences in terminal nodes (or other downstream internal nodes).
>
> Finally, you ask the crucial question whether our promising experimental results constitute evidence that our inductive biases are correct in practice. Our results on semi-synthetic datasets can only provide indirect evidence, in that our approaches perform well on simulated benchmarks that have been accepted by the community as useful for testing algorithms. Therefore, we consider the results on the real dataset Twins most promising in this regard, as they show actual evidence that these inductive biases are useful in a real-world setting.
>
>
> __*(3) Possible variants on the reparametrization approach.*__
> Your proposed variation on the reparametrization approach is a very interesting idea, which  could actually conceptually be seen as a simplified version of FlexTENet, in that it allows to learn in a data-driven manner how to best parametrize an *additive* PO relationship. To implement the same idea in an end-to-end manner using FlexTENet, one would simply have to remove all internal forward passes from shared to private layers (corresponding to a removal of all internal diagonal arrows in Fig 2.3), resulting in three independent networks (one shared, two $\mu_w$-specific). In the experimental setting B, where the treatment effect is indeed additive, we would expect this simplified version to match the performance of FlexTENet. For DGPs where the shared structure is not additive, as in IHDP, this proposed version would not overcome the problem of incorrect parametrization due to a decrease in flexibility, so we would not expect it to match the performance of FlexTENet.
>
> Further, you ask whether (variants of) the reparametrization approach would fail when the $\mu_w$ are of comparable complexity. We would argue that the answer to this question would depend on the complexity of the treatment effect/ the amount of shared structure between the $\mu_w$. In one extreme, if $\mu_1(x)$ and $\mu_0(x)$ are of comparable complexity because \tau(x)=c (constant), the reparametrization approach (and all other approaches studied) would do very well (as seen in the experiments A & B when $\rho=0$). If, on the other hand, $\mu_1(x)$ and $\mu_0(x)$ are of comparable complexity but completely different (e.g. depending on disjoint feature sets), then $\tau(x)$ would be most complex, and the reparametrization approach would fail. As discussed above, we would not consider the latter a particularly realistic scenario in practice.

---

> > ### Comment · Reviewer_9oJ5 · 2021-08-10
> > **Thank you**
> >
> > Thank you for your detailed answers to my questions.

---

### Official Review · Reviewer_yc6v · 2021-07-14

**Rating:** 6
**Confidence:** 3

**Summary:**

This paper considers the problem of estimating the conditional average treatment effect (CATE) of a binary treatment with some potential outcomes of interest (either continuous or binary). Simply put, in, e.g., a medical scenario, CATE would measure the expected delta in administering some drug vs. not for a particular patient X. The main challenge of this problem is that Y(0) and Y(1), the outcomes given the value of the binary treatment, are never observed at the same time, so estimating CATE is not a standard supervised learning problem, where Y(1) - Y(0) can be observed for the same input X. One can estimate both Y(1) and Y(0) separately, but the errors add.

The key insight of this work is that the difference, $\tau(x) = \mathbb{E}[Y(1) - Y(0) \mid X = x]$, is often of a much simpler form than either $\mathbb{E}[Y(1) \mid X = x]$ or $\mathbb{E}[Y(0) \mid X = x]$, due to inherent similarities in Y(1) and Y(0). To account for this, this paper investigates and compares several end-to-end learning strategies that encode inductive bias for shared/similar structure---namely, (1) regularization, (2) reparameterization, and (3) shared multi-task architectures.

**Limitations And Societal Impact:**

I do not foresee any major limitations and/or societal impacts. How to choose regularization strengths would be the main implementation question/limitation.

**Main Review:**

=== Strengths ===

- The paper is very well written and provides extensive evaluation and discussion/comparison of past work.
- In general, the problem that the paper focuses on is interesting and important.
- It is nice to see *some* real world evaluation (Twins dataset), with the understanding that this is imperfect and hard to come by.
- Empirically, the strategies proposed by the authors (in particular, FlexTENet) seem to show consistent improvements over competing baselines.
- The proposed strategies are simple and easy to implement within most frameworks, and should be easy to adopt.

=== Weaknesses ===

- From a technical standpoint, the contributions of this paper are fairly limited. The inductive biases for shared/similar structure are, for the most part, well-known (at least in the multi-task learning community). This is not inherently bad---there remains value in the analysis. The wording of lines 61-70 is a bit confusing though, as shared structure inductive bias has already been a quite common "dimension" for improving MTL estimators (not just the architectures used). The use in CATE estimation specifically may be a bit less prevalent.

- While it may be true that this shared structure inductive bias may be useful, it seems hard to systematically tune the regularization strengths, $\lambda$. This is especially true in the real world, where development data will be mostly unavailable, and the true underlying setting (shared structure or not) will be unknown. In the worst case, it seems that strategies (1) and (3) with too high of a regularization could severely underestimate CATE (i.e., the models collapse together)? How should these weights be chosen?

=== Justification for score ===

Overall, the paper is well-written and provides an interesting survey of several reasonable inductive-bias inducing techniques for CATE estimation. The technical novelty of the paper, however, is fairly limited, and its experimental findings are not that surprising given what we know from MTL literature, which makes it borderline. That said, its presentation and analysis could still make a potential contribution to the treatment effect estimation community.

=== Minor comments ===
- It would be helpful to introduce the CATE estimation problem more formally (as done in Section 2) earlier in the introduction. This would help ground readers less familiar with PO frameworks.

**Time Spent Reviewing:**

7

---

> ### Author Response · Authors · 2021-08-10
> **Response to Reviewer yc6v**
>
> Thank you for your thoughtful comments and suggestions. We are very grateful for all your appreciative comments regarding our writing, problem setting, applicability and empirical evaluation. Below, we give answers to your two major comments in turn.
>
> ***
> __*(A) Contributions of the paper*__
>
> We agree that the *technical* contributions of this paper are limited relative to existing work in *multi-task learning (MTL)*: In fact, as we state in our opening paragraph, it was indeed our goal “To further advance the understanding of how to incorporate insights from other areas of ML into treatment effect estimation”. We are glad that you share our opinion that this is not ‘inherently bad’ and hope to further convince you of the novelty and value of our contributions to the CATE estimation literature through the points we make below!
>
> 1. *The connection between our research question and MTL is non-trivial.* As we discussed in l.47-58 and l.118-133, the original motivation behind this work was the excellent performance of the multi-stage model-agnostic ‘meta-learner’ strategies proposed recently in papers with more statistical focus [7, 8, 18, 19], which benefit from targeting CATE directly in a second-stage estimation step. We wanted to investigate whether it would be possible to design end-to-end solutions (in true ‘ML-style’) to match or improve upon their performance. From this standpoint, it was not obvious that techniques from MTL, which exploit shared structure in outcomes, could be used to match the state-of-the-art in statistical ML for CATE. Conversely, the state-of-the-art ML literature on end-to-end learning of POs had not considered targeting CATE *directly*. We therefore consider the connection that we made between the literatures on (i) shared structure inductive bias in MTL and (ii) CATE estimation non-trivial.
>
> 2. *The results of transferring techniques from MTL to CATE estimation were non-obvious.* We would like to argue that it was not obvious whether techniques that have been successful in the MTL context would perform well when applied to CATE estimation. This is because (as we state in l.271f) while MTL is concerned with achieving a good *average performance in prediction of outcomes* across tasks, the main target of CATE estimation is estimating the *difference between outcomes* – which is a completely different statistical target parameter and goal.   We do agree, however, that it is not surprising that MTL techniques would help in estimation of the POs separately (which can be seen as an example of two related regression tasks). We will try to make this distinction between the relationship of MTL with (i) PO estimation and (ii) CATE estimation more clear.
>
> Finally, allow us to emphasize that we did not aim to make any contributions to the *MTL literature*, nor did we intend to imply that we do in the passage you mention (l. 61-70); in fact, we had attempted to carefully articulate that our goal was to explore and improve *CATE estimators*. As we tried to highlight above, our main connection to MTL is that this literature happened to have developed the tools that we needed to answer our research question in the CATE estimation context. We will try to make this even more clear in the updated manuscript.  We would also like to  argue that our novelty and contributions, while not of algorithmic nature, go beyond providing a mere ‘survey’: we think that they lie in introducing and analyzing a new and important modeling dimension (for CATE estimators), providing actionable insights that are orthogonal to what is contained in most of the existing ML literature on CATE estimation.
>
> __*(B)	Hyperparameter tuning*__
>
> Thank you for pointing out that this important point warrants discussion in our paper. We will use the additional space provided in the final manuscript to include a discussion of the below in the updated paper.
>
> It is indeed a well-known problem in the CATE estimation literature that model selection is nontrivial due to the absence of counterfactuals in practice. The testable implications of the shared structure bias, however, are different for (i) the PO estimation and (ii) the CATE estimation problems, which is a feature that we would suggest to exploit in choosing hyperparameter settings. That is, while the usefulness of the inductive biases for estimation *of CATE* cannot easily be verified,  their usefulness for estimating the*the POs* can be verified through cross-validation on held-out factual observations.
>
> Unfortunately, good performance on estimation of the POs is not sufficient. To see this, note that the tuples of PO estimates $(\mu_0(x)+e(x), \mu_1(x)+e(x))$ and $(\mu_0(x)-e(x), \mu_1(x)+e(x))$ [where $e(x)$ is an error and $\mu_w(x)$ is the ground truth] will in expectation give exactly the same MSE when evaluated using held-out factual observations. Yet, the first tuple will make no error in estimating CATE, while the second tuple makes an error of $2e(x)$. This highlights that error can either compound or cancel across the POs, therefore making it possible that a hyperparameter setting resulting in better fit on the POs also results in worse fit on CATE, and vice versa. Because an estimators’ performance on estimating CATE remains unobservable in practice, a good inductive bias is needed to choose between models that have equal predictive performance on POs. As we discuss in the paper, we consider preferring shared structure (or a simple CATE) a reasonable choice for this in many practical applications.
>
> *Proposed approach for setting hyperparameters.* As a simple heuristic to set such hyperparameters in practice, we would thus recommend the following scheme that trades off between imposing an assumption (“CATE is most likely simple”) and factual performance:
> 1)	Start with $\lambda_2$ small, and keep increasing it while held-out predictive performance does not decrease.
> 2)	Set $\lambda_2$ to its largest value for which predictive performance did not deteriorate.
>
> In a sufficiently flexible (overparameterized) model class in which multiple PO estimators induce the same empirical performance, such a scheme allows to pick the hyperparameter setting resulting in the least complex CATE while remaining compatible with factual observations.
>
> *Illustrative results* To illustrate the soundness of this approach, we present results on Setting B (with $n_0=2000$ and $n_1=500$), and consider tuning $\lambda_2$ of FlexTENet for different values of $\rho$ (recall that as $\rho$ increases, the degree of shared structure decreases).
>
> Table 1: Factual RMSE versus RMSE($\tau(x)$) of FlexTENet with different values of $\lambda_2$ for setting B across $\rho$, evaluated on a hold-out set. Averaged across 10 replications, standard error in parentheses.
>
> |                     | Factual RMSE $\rho=0$ | RMSE($\tau(x)$) $\rho=0$ | Factual RMSE $\rho=0.2$ | RMSE($\tau(x)$) $\rho=0.2$ | Factual RMSE $\rho=0.8$ | RMSE($\tau(x)$) $\rho=0.8$ |
> |---------------------|-----------------------|--------------------------|-------------------------|----------------------------|-------------------------|----------------------------|
> | $\lambda_2=10^{-4}$ | 1.131 (.017)          | 0.513 (.016)             | 1.129 (.016)            | 0.557 (.013)               | **1.113** (.015)          | 0.622 (.016)               |
> | $\lambda_2=10^{-3}$ | 1.131 (.017)          | 0.507 (.014)             | 1.129 (.016)            | 0.552 (.014)               | 1.115 (.015)            | 0.619 (.016)               |
> | $\lambda_2=10^{-2}$ | 1.126 (.016)          | 0.426 (.012)             | **1.126** (.016)          | **0.523** (.012)             | 1.118 (.016)            | **0.608** (.015)             |
> | $\lambda_2=10^{-1}$ | **1.112** (.017)        | **0.189** (.010)           | 1.134 (.016)            | 0.547 (.022)               | 1.132 (.016)            | 0.700 (.022)              |
>
>
> We observe that, as expected, the effect of $\lambda_2$ on factual performance is much smaller than on estimation of CATE (note that the factual outcomes include noise with standard deviation 1, which explains the larger magnitude in factual RMSE). Following the heuristic of choosing the largest $\lambda_2$ without performance decrease would lead to the best results on two of the three considered settings; only for $\rho=0.8$, $10^{-2}$ would be better than the chosen $10^{-4}$, but not significantly.
>
> We will present an extended empirical analysis of the effect of $\lambda_2$ and the relationship between factual performance and CATE estimation quality in the appendix.

---

> > ### Comment · Reviewer_yc6v · 2021-08-20
> > **Review response**
> >
> > Thank you for your helpful answers to my questions.
> >
> > The additional hyperparameter tuning study is good to include. I also can see that there are non-trivial components to applying MTL techniques to CATE. I am not quite familiar enough with the specific research subdomain (effect estimation) as to argue on how impactful these ideas on this area (as opposed to more general MTL), but it is a well written, clear, and well analyzed paper which I think would serve as a useful contribution. I am therefore raising my score.

---

> ### Author Response · Authors · 2021-08-18
> **Dear Reviewer yc6v**
>
> Once more, we would like to thank you for your invaluable feedback! We were wondering whether our response (Aug 10) has addressed your concerns. Should you have any leftover comments or concerns, please let us know - we would be happy to do our utmost to address them!

---

### Official Review · Reviewer_EVDi · 2021-07-16

**Rating:** 6
**Confidence:** 4

**Summary:**

In this paper, the authors proposed methods to encourage structural similarities in the potential outcomes (POs) under different treatments when learning the conditional average treatment effects (CATEs). Specifically, three sets of different structure/model modifications were proposed on top of the existing TNet/TarNet model by utilizing regularizations or shared hidden units under different treatments. Proposed methods were evaluated on various simulated data.

**Limitations And Societal Impact:**

The authors should discuss more on the limitation and societal impact when the inductive bias for encouraging similarity between POs is not correct. How much would this impact the estimation and treatment recommendation should be evaluated. Potential remedies and proper guidance on whether this line of methods should be utilized should be discussed.

**Main Review:**

The proposed methods are structure modifications to existing methods for learning CATEs. The modifications are fairly straightforward, utilizing regularization or shared hidden units in neural networks. These techniques are commonly used in neural network models, so the contribution mainly lies in the application area for learning CATEs under the inductive bias assumption.

The proposed methods are valid and appropriate to address the research question. This paper is in general well organized. Extensive simulation studies were conducted. If the true POs share a common structure, the proposed methods are useful in improving the estimation for CATEs by borrowing strengths and encouraging a simpler model structure. My main concern is the soundness of the assumption and how the proposed models perform under violation of the assumptions. Also, how to test the assumption and whether people should proceed with this line of methods when there is no prior knowledge of POs need to be examined.

Additional comments and questions:

In the simulations, all settings assumed some common structure between the two POs. As a sensitivity analysis, I wonder how would the models perform if the two POs do not share any structure or parameter? In this circumstance, would the penalization parameter that encourages similarity to be chosen close to 0 by cross-validation?

Can the authors talk about how would people check the assumptions that the POs are similar, and whether or not to encourage a similar structure between POs in learning CATEs when there is no prior knowledge?

Also, I wonder do the authors compare the accuracy in inferring the optimal treatments under different methods? MSEs of CATEs were reported. But sometimes a more meaningful question is whether the treatment recommendation direction is correct, and such result can be informative to viewers.


**Time Spent Reviewing:**

5

---

> ### Author Response · Authors · 2021-08-10
> **Response to Reviewer EVDi: [Part 2/2] Sensitivity Analyses and the Impact of Incorrect Inductive Biases**
>
> __*(C) Sensitivity analysis: What if there is no shared structure?*__
>
> We agree that sensitivity analyses are important; this is precisely the reason why we included the experimental knob $\rho$ in our simulation settings. In setting B, when $\rho=0.8$, only 20\% of terms appear in both regression surfaces – i.e. there is little shared structure – presenting a substantial reduction relative to 100\% shared structure when $\rho=0$.
>
> As we have highlighted in Table 1 of our response to point (B) above, if hyperparameters are chosen based on factual performance, the chosen parameter $\lambda_2$ will indeed decrease in magnitude when the degree of shared structure vanishes.
>
> In our paper, we did not present a setting where there is no shared structure at all, as we could not come up with an example where we would consider this realistic in practice. Nonetheless, to answer your question, in Table 2 below we ran an experiment using the same data and outcome specification as in settings A & B in the paper, but now used only the linear terms in $\mu_0(x)$ and only the nonlinear terms in $\mu_1(x)$ (so that there is no shared structure between the surfaces.)
>
> Table 2: Factual RMSE versus RMSE($\tau(x)$) for a setting without shared structure ($n_0=n_1=2000$) evaluated on a hold-out set. Averaged across 10 replications, standard error in parentheses.
>
>
> |                                | RMSE($\tau(x)$) | Factual RMSE  |
> |--------------------------------|-----------------|---------------|
> | TNet                           | 0.498 (.017)   | 1.065 (.012) |
> | DRNet                          | 0.528 (.018)   | n/a           |
> | FlexTENet ($\lambda_2=10^{-1}$) | 0.592 (.014)   | 1.086 (.013) |
> | FlexTENet ($\lambda_2=10^{-2}$) | 0.506 (.015)   | 1.072 (.011) |
> | FlexTENet ($\lambda_2=10^{-3}$) | 0.501 (.014)   | 1.062 (.010)  |
>
> As expected, TNet performs best on this setup. Due to its flexibility, FlexTENet almost matches the performance of TNet when $\lambda_2$ is not set too high. Further, in line with expectations, smaller values for $\lambda_2$ would be chosen based on factual evaluation.
>
> __*(D) Societal impact: What if the inductive bias is not correct?*__
>
> Thank you for raising an interesting and important question regarding the societal implications of incorrect inductive biases.
>
> First, we would like to emphasize once more that existing learning strategies are not free of inductive biases (as discussed in Sec. 4.1), implying that no approach will *always* lead to correct conclusions. In essence, your question therefore boils down to the following: Is it better (i) to discover only little TE heterogeneity when potentially more exists or (ii) to discover more heterogeneity when little or none exists? We consider this a complex normative question, whose answer will necessarily depend on applications and their stakes, and should thus be assessed by domain experts in practice.
>
> Nonetheless, we would argue that our rather conservative approach (i.e. (i)) is at least most in line with how policy/treatment decisions are currently made in practice, where the focus still lays mainly on the *average* treatment effect (corresponding to enforcing *no* effect heterogeneity). Further, we would argue that it is generally safer to impose the assumption of less heterogeneity when little prior knowledge is available, as this (conceptually) results in more statistical strength being shared across individuals. That being said, relying on average treatment effects may disproportionally affect minority groups, which is another ethical concern to consider.

---

> > ### Comment · Reviewer_EVDi · 2021-08-21
> > **Review response**
> >
> > Thank the authors for the detailed response. I think the assumption of the shared structure is well discussed in the response. It would be good to add them to the paper to better justify the proposed methods. Because of the gap between accurately estimating POs and CATEs, the proposed approach for choosing the tuning parameter may not lead to the best model, as demonstrated in your analyses. I think this should be clearly pointed out in the paper to inform the readers. Overall, my main concerns are well addressed.

---

> ### Author Response · Authors · 2021-08-10
> **Response to Reviewer EVDi: [Part 1/2] Soundness of Assumptions and Model Selection in Practice**
>
> Thank you for your thoughtful comments and suggestions. We give answers to each in turn: Sections (A – C) below are aimed at the “main concerns” raised in the main review and Section (D) aims at Limitations and Societal Impact.
> ***
>
> __*(A) Soundness/realism of assumption and resulting inductive biases*__
>
> We agree that it is important to justify the soundness of assumptions and resulting inductive biases, and had attempted to do so at multiple places in the paper. Based on your comments, we now realize that our paper may benefit from a more centralized discussion of the arguments detailed below, which we will include in the updated submission.
>
> Our central assumption and resulting inductive bias is ‘there is shared structure between the POs’ (see .e.g. Sec. 4.2.1). Below, we detail six arguments why we consider this a reasonable and useful assumption and/or inductive bias within the CATE estimation context:
>
> 1. *Shared structure is a reasonable assumption in many practical applications (e.g l.37-42).* In all practical applications we are familiar with (mainly in medicine & economics), one would expect at least *some* similarities between treated and untreated individuals: intuitively speaking, receiving a drug will most likely not change *all* biological processes related to a disease progression in a patient, and attending a job training program is unlikely to neutralize *all* characteristics determining an individual’s salary. In medicine, for example, this has led to the well-known distinction between prognostic and predictive (effect-modifying) biomarkers [16, 17]; in our context, the strength of such prognostic information would determine the degree of shared structure.
> 2. *Assuming shared structure is compatible with explicit assumptions made in (recent) theoretical work (e.g. l.42 -46).* Multiple recent papers on CATE meta-learners from the statistics community [7, 18, 19], make the assumption that CATE is a simpler function (e.g. smoother, sparser) than each of the POs – which implies shared structure (i.e. a shared baseline function) between the POs. Additionally, a related (but much stronger) assumption is made in papers considering the popular semi-parametric “Partially linear regression model” analyzed in e.g. [44, 45]; here all nonparametric (‘complex’) structure is shared between POs, while the treatment effect is assumed parametric (often constant).
> 3. *Imposing our inductive biases conceptually corresponds to collecting evidence against a scientific null hypothesis of no treatment effect heterogeneity (e.g. l.35).* By preferring shared structure across the POs a priori, our inductive biases ensure that effect heterogeneity is discovered only if there is sufficient evidence for it in the data. Especially in scenarios where it is unknown whether a treatment has a (heterogeneous) effect at all, this conceptually corresponds to collecting evidence against the null hypothesis we consider most natural in this context. Note that the implicit inductive biases in existing learners (Sec. 4.1) do not even allow to reason in terms of a null hypothesis on the target parameter $\tau(x)$– they correspond to null hypotheses on the POs separately.
> 4. *We create an explicit inductive bias on the target parameter, which was previously lacking (e.g. l.197f).* The implicit inductive biases in existing learners (Sec. 4.1) target the nuisance parameters $\mu_w(x)$ instead of the true target parameter $\tau(x)$, making it impossible for an investigator to control the implied complexity of $\tau(x)$ – our approaches make this possible. In a Bayesian context, a similar argument in favor of placing priors over $\tau(x)$ instead of the $\mu_w(x)$ is made in [25].
> 5. *Shared structure is implicitly assumed in virtually all (semi)synthetic DGPs used in related work.* Most of our references which simulate their own PO functions in experiments assume that there is a shared baseline outcome and an additive CATE in their DGPs, e.g. [2, 7, 8, 14, 17, 18, 19, 24]. Further, in [1, 13] the POs also have shared structure but it is not additive. Only [8, 19] include a simulation setting without any shared structure for illustrative purposes, described by the authors as ‘unusual in practice’[8] and ‘highly stylized’ [19].
> 6. *We provide ample empirical evidence for the assumption’s usefulness on both popular semi-synthetic and a real-world data set.* We empirically confirm that our inductive biases are useful not only across widely used semi-synthetic benchmark data sets (IHDP [1] and ACIC2016[43]), but also on the real dataset Twins (which is the only fully real dataset we are aware of).
>
> Additional references, with numbering continued from main manuscript:
>
> [44] Robinson, P. M. (1988). Root-N-consistent semiparametric regression.
>
> [45] Chernozhukov, V., Chetverikov, D., Demirer, M., Duflo, E., Hansen, C., Newey, W., & Robins, J. (2018). Double/debiased machine learning for treatment and structural parameters.
> ***
> __*(B) Testability of assumptions and how to choose hyperparameters in practice*__
>
> Thank you for pointing out that this important point warrants discussion in our paper. We will use the additional space provided in the final manuscript to include a discussion of the below in the updated paper.
>
> *Testability.* Our assumptions have both testable and untestable implications: the usefulness of the inductive biases for estimation *of the POs* can be verified through cross-validation, while the usefulness for estimation *of CATE* cannot easily be verified. That is (as we show in our experiments) adding our inductive biases can improve upon estimation of the POs – which can be verified in practice by performing cross-validation on held-out factual observations. Due to the absence of the counterfactual difference $Y(1)-Y(0)$ in practice, however, it is generally not possible to cross-validate hyperparameters like $\lambda_2$ to assess their expected performance in CATE estimation.
>
> *Illustration of the testability issue.* To see this, note that the tuples of PO estimates $(\mu_0(x)+e(x), \mu_1(x)+e(x))$ and $(\mu_0(x)-e(x), \mu_1(x)+e(x))$ [where $e(x)$ is an error and $\mu_w(x)$ is the ground truth] will in expectation give exactly the same MSE when evaluated using held-out factual observations. Yet, the first tuple will make no error in estimating CATE, while the second tuple makes an error of $2e(x)$. This highlights that errors can either compound or cancel across the POs, therefore making it possible that a hyperparameter setting resulting in better fit on the POs also results in worse fit on CATE, and vice versa. Unfortunately, an estimators’ performance on estimating CATE remains unobservable in practice.
>
> *Proposed approach for setting hyperparameters:* As a simple heuristic to set such hyperparameters in practice, we would thus recommend the following scheme that trades off between imposing an assumption (“CATE is most likely simple”) and factual performance:
> 1)	Start with $\lambda_2$ small, and keep increasing it while held-out predictive performance does not decrease.
> 2)	Set $\lambda_2$ to its largest value for which predictive performance did not deteriorate.
>
> In a sufficiently flexible (overparameterized) model class in which multiple PO estimators induce the same empirical performance, such a scheme allows to pick the hyperparameter setting resulting in the least complex CATE while remaining compatible with factual observations.
>
> *Illustrative results:* To illustrate the soundness of this approach, we present results on Setting B (with $n_0=2000$ and $n_1=500$), and consider tuning $\lambda_2$ of FlexTENet for different values of $\rho$ (recall that as $\rho$ increases, the degree of shared structure decreases).
>
> Table 1: Factual RMSE versus RMSE($\tau(x)$) of FlexTENet with different values of $\lambda_2$ for setting B across $\rho$, evaluated on a hold-out set. Averaged across 10 replications, standard error in parentheses.
>
> |                     | Factual RMSE [$\rho=0$] | RMSE($\tau(x)$) [$\rho=0$] | Factual RMSE [$\rho=0.2$] | RMSE($\tau(x)$) [$\rho=0.2$] | Factual RMSE [$\rho=0.8$] | RMSE($\tau(x)$) [$\rho=0.8$] |
> |---------------------|-----------------------|--------------------------|-------------------------|----------------------------|-------------------------|----------------------------|
> | $\lambda_2=10^{-4}$ | 1.131 (.017)          | 0.513 (.016)             | 1.129 (.016)            | 0.557 (.013)               | **1.113** (.015)          | 0.622 (.016)               |
> | $\lambda_2=10^{-3}$ | 1.131 (.017)          | 0.507 (.014)             | 1.129 (.016)            | 0.552 (.014)               | 1.115 (.015)            | 0.619 (.016)               |
> | $\lambda_2=10^{-2}$ | 1.126 (.016)          | 0.426 (.012)             | **1.126** (.016)          | **0.523** (.012)             | 1.118 (.016)            | **0.608** (.015)             |
> | $\lambda_2=10^{-1}$ | **1.112** (.017)        | **0.189** (.010)           | 1.134 (.016)            | 0.547 (.022)               | 1.132 (.016)            | 0.700 (.022)              |
>
> We observe that, as expected, the effect of $\lambda_2$ on factual performance is much smaller than on estimation of CATE (note that the factual outcomes include noise with standard deviation 1, which explains the larger magnitude in factual RMSE). Following the heuristic of choosing the largest $\lambda_2$ without performance decrease would lead to the best results on two of the three considered settings; only for $\rho=0.8$, $10^{-2}$ would be better than the chosen $10^{-4}$, but not significantly.
>
> We will present an extended empirical analysis of the effect of $\lambda_2$ and the relationship between factual performance and CATE estimation quality in the appendix.

---

> ### Author Response · Authors · 2021-08-18
> **Dear Reviewer EVDi**
>
> Once more, we would like to thank you for your invaluable feedback! We were wondering whether our response (Aug 10) has addressed your concerns. Should you have any leftover comments or concerns, please let us know - we would be happy to do our utmost to address them!

---

### Decision · Program_Chairs · 2021-09-27

**Decision:**

Accept (Spotlight)

**Comment:**

This paper presents a novel framing and approach to the problem of conditional average treatment effect estimation, and validates it on synthetic data and a small sample of natural data. Reviewers generally found it to be a technically-simple but interesting contribution, and raised only minor concerns.